# GUMBEL COUNTERFACTUAL GENERATION FROM LANGUAGE MODELS

**Shauli Ravfogel**[1]* **Anej Svete**[2]* **Vésteinn Snæbjarnarson**[2,3] **Ryan Cotterell**[2]
[1]New York University  [2]ETH Zurich  [3]University of Copenhagen
{shauli.ravfogel, vesteinnsnaebjarnarson}@gmail.com
{anej.svete, ryan.cotterell}@inf.ethz.ch

## ABSTRACT

Understanding and manipulating the causal generation mechanisms in language models is essential for controlling their behavior. Previous work has primarily relied on techniques such as representation surgery—e.g., model ablations or manipulation of linear subspaces tied to specific concepts—to *intervene* on these models. To understand the impact of interventions precisely, it is useful to examine *counterfactuals*—e.g., how a given sentence would have appeared had it been generated by the model following a specific intervention. We highlight that counterfactual reasoning is conceptually distinct from interventions, as articulated in Pearl's causal hierarchy. Based on this observation, we propose a framework for generating true string counterfactuals by reformulating language models as a structural equation model using the Gumbel-max trick, which we called Gumbel counterfactual generation. This reformulation allows us to model the joint distribution over original strings and their counterfactuals resulting from the same instantiation of the sampling noise. We develop an algorithm based on hindsight Gumbel sampling that allows us to infer the latent noise variables and generate counterfactuals of observed strings. Our experiments demonstrate that the approach produces meaningful counterfactuals while at the same time showing that commonly used intervention techniques have considerable undesired side effects.

https://github.com/shauli-ravfogel/lm-counterfactuals

## 1 INTRODUCTION

The study of language model (LM) interpretability often borrows terminology from Pearl's causal calculus (Pearl, 1989), e.g., researchers often talk of *intervening* on a model's parameters and *counterfactually* generating strings. Pearl's framework distinguishes between three levels of causal reasoning (Shpitser & Pearl, 2008). Association, the first level, pertains to statistical correlations, i.e., observing patterns observed in data without interacting with the world. Intervention, the second level, pertains to actively changing variables in the world and observing their effects at a macro level. Counterfactuality, the third level, pertains to imagining what could have happened if past events had unfolded differently. However, LM literature often uses these three causal terms casually and at times imprecisely—particularly when it comes to counterfactuality, which remains challenging to rigorously define (Feder et al., 2022; Mueller, 2024; Mueller et al., 2024). In this paper, we apply a well-defined notion of counterfactuality in LMs using the framework of structural equation modeling.

Efforts to exert control over LMs have led to substantial research on targeted *interventions* in the models. One such technique is *representation surgery*, which involves modifying an LM's architecture to manipulate its internal representation space (Lakretz et al., 2019; Vig et al., 2020; Feder et al., 2021; Ravfogel et al., 2021b; Elhage et al., 2021; Elazar et al., 2021; Nanda, 2023; Syed et al., 2023; Kramár et al., 2024; Avitan et al., 2024). The *linear subspace hypothesis* (Bolukbasi et al., 2016; Vargas & Cotterell, 2020; Ravfogel et al., 2022) posits that human-interpretable *concepts*, such as gender or grammatical number, are encoded within specific linear subspaces of the LM's representation space. This makes it possible to perform precise interventions on these high-level concepts, such as removing the concept's information by projecting the representations onto the

---

*  Equal contribution.

complement of the concept subspace (Ravfogel et al., 2020; 2021a; 2022; 2023; Guerner et al., 2024; Scalena et al., 2024; Singh et al., 2024). These interventions modify the model and allow researchers to examine its behavior after the change. However, while interventions can induce a *change* in the model, they cannot answer counterfactual questions, e.g., what would a given string look like if it had been generated by the model after the intervention?

Counterfactual analysis, as defined by Pearl, is challenging because it requires describing the system of interest through a causal model that enables counterfactual reasoning. In this paper, we address this challenge for LMs by turning to structural equation models (SEMs; Wright, 1921; Haavelmo, 1944). SEMs break down a probabilistic model into *exogenous* variables, which account for latent randomness, and *endogenous* variables, which are deterministic once the exogenous ones are fixed. To frame LMs as SEMs, we turn to the Gumbel-max trick (Gumbel, 1954), which separates the deterministic computation of next-symbol logits from the sampling process. This approach has been explored in reinforcement learning (Oberst & Sontag, 2019), but has not yet been considered in the language modeling domain. Indeed, we discuss later, the infinite outcome space of LMs requires special attention that is not needed in finite domains. Additionally, we highlight that the SEM derived from the Gumbel parameterization of an LM is not unique—while some parameterizations are unnatural (Oberst & Sontag, 2019, §3.1), there are several natural choices to model an LM causally. We argue for Gumbel counterfactual generation by appealing to the classic Thurstone discriminal process (Thurstone, 1927),[1] and because its simplicity paves the way for other choices beyond Gumbel noise (Lorberbom et al., 2021; Haugh & Singal, 2023).

Our formulation allows for generating counterfactual strings by sampling from conditional noise distributions, enabling precise analysis of string-level effects from interventions in models like GPT2-XL (Radford et al., 2018) and LLaMA3-8b (Touvron et al., 2023). Despite targeting specific behaviors through interventions such as linear steering (Li et al., 2024; Singh et al., 2024), knowledge editing (Meng et al., 2023), and instruction tuning (Wei et al., 2022), results reveal unintended side effects, e.g., gender-based interventions unexpectedly altering unrelated completions. These findings challenge the goal of achieving *minimal change* and show that even localized parameter modifications can have broader, undesired impacts.

## 2 LANGUAGE MODELS AS STRUCTURAL EQUATION MODELS

Let $\Sigma$ be an **alphabet**—a finite, non-empty set of symbols. A **language model** (LM) is a probability distribution over $\Sigma^*$, the set of all strings formed from symbols in $\Sigma$. A **language encoder** is a function $\boldsymbol{h_\theta} \colon \Sigma^* \to \mathbb{R}^d$ parameterized by $\boldsymbol{\theta}$ that maps strings to $d$-dimensional vectors (Chan et al., 2024). Representational surgery is performed by intervening on $\boldsymbol{h_\theta}$. Popular architectures for implementing language encoders include Transformers (Vaswani et al., 2017) and RNNs (Elman, 1990). Language encoders are particularly valuable because under mild conditions (Du et al., 2023, Thm. 4.7), they ensure the model defines a distribution over strings—thus inducing an LM—as

$$p(\boldsymbol{w}) = p(w_1 \cdots w_T) \tag{1a}$$

$$= p(\text{EOS} \mid \boldsymbol{w}) \prod_{t=1}^{T} p(w_t \mid \boldsymbol{w}_{<t}) \tag{1b}$$

$$= \text{softmax}(\mathbf{E}\,\boldsymbol{h_\theta}(\boldsymbol{w}) + \mathbf{b})_{\text{EOS}} \prod_{t=1}^{T} \text{softmax}(\mathbf{E}\,\boldsymbol{h_\theta}(\boldsymbol{w}_{<t}) + \mathbf{b})_{w_t}. \tag{1c}$$

Here, $\mathbf{E} \in \mathbb{R}^{|\overline{\Sigma}| \times d}$ and $\mathbf{b} \in \mathbb{R}^{|\overline{\Sigma}|}$ is a bias term. We assume that EOS $\notin \Sigma$ and define $\overline{\Sigma} \overset{\text{def}}{=} \Sigma \cup \{\text{EOS}\}$. The quantity $\ell\left(\overline{w}; \boldsymbol{w}_{<t}\right) \overset{\text{def}}{=} (\mathbf{E}\,\boldsymbol{h_\theta}(\boldsymbol{w}_{<t}) + \mathbf{b})_{\overline{w}}$ is often termed the **logit** of $\overline{w}$. We refer to LMs of the form Eq. (1c) **representational LMs**.

---

[1] Assuming the Thurstonian model of decision-making, equivalent to Luce's axiom of choice (Luce, 1959), makes the causal process behind an LM *identifiable* and is identical to having been generated through Gumbel noise. However, our appeal to Thurstone is most concretely an appeal to precedent, aligning with its established role in modeling decision-making and preference aggregation in previous work.

## 2.1 STRUCTURAL EQUATION MODELING

We briefly review SEMs, which provide a framework for discussing causal manipulations of a generation process and allow us to define the intuitive notion of a counterfactual precisely. See Pearl (2009), Pearl et al. (2016), and Peters et al. (2017) for a more in-depth treatment. We offer a definition of an *acyclic* SEM because our focus is on language modeling, which operates on countably infinite random variables (RVs) with no cyclic dependencies. To apply acyclic SEMs to language modeling, we *expose* the acyclicity by specifying a partial order $\prec$ that characterizes the acyclic relationships. This establishes a common interface for both the finite and infinite-variable models.

**Definition 2.1** (Acyclic SEM). *An **acyclic SEM (ASEM)** is a five-tuple $\mathcal{M} = (\mathbf{V}, \mathbf{U}, \mathcal{P}, \mathcal{F}, \prec)$, where $\mathbf{V} = \{V_1, \ldots, V_N\}$ is a finite set of **endogenous RVs**, $\mathbf{U} = \{U_1, \ldots, U_N\}$ is a finite set of **exogenous RVs**, $\mathcal{P} = \{\mathbb{P}_1, \ldots, \mathbb{P}_N\}$ is a set of probability distributions where $\mathbb{P}_n$ is a distribution over $U_n \in \mathbf{U}$, $\mathcal{F} = \{f_1, \ldots, f_N\}$ is a finite set of **structural equations**, and $\prec$ is a **partial order** on $\mathbf{V}$. We require that the endogenous RVs satisfy the structural equations, i.e., we require that*

$$V_n = f_n(\mathrm{PA}(V_n), U_n), \tag{2}$$

*for $n \in \{1, \ldots, N\}$, where $\mathrm{PA}(V_n)$, the subset of $\mathbf{V}$ in the domain of $f_n$, respects $\prec$, i.e., we require that $\mathrm{PA}(V_n) \subseteq \{V \in \mathbf{V} \mid V \prec V_n\}$.[2,3]*

$\mathcal{P}$ induces a joint probability distribution over $\mathbf{U}$; together with $\mathcal{F}$, it also induces a unique distribution over $\mathbf{V} \cup \mathbf{U}$ due to the acyclicity. Given an outcome $\mathbf{U} = \mathbf{u}$, a **solution** of an ASEM is any assignment of the variables $\mathbf{V} = \mathbf{v}$ that satisfies the equations in $\mathcal{F}$ putting $\mathbf{U} = \mathbf{u}$. In ASEMs, any outcome $\mathbf{U} = \mathbf{u}$ induces a unique solution (Peters et al., 2017); in the cyclic case, multiple valid assignments can exist (Peters et al., 2017, Problem 3.8).

An ASEM can also be viewed as a directed graph over vertices $\mathbf{U} \cup \mathbf{V}$ with a directed edge from every $X \in \mathrm{PA}(V_n) \cup \{U_n\}$ to $V_n$. In it, the exogenous variables always appear without parents. This presentation allows one to visualize ASEMs compactly and, also, reason better about their properties (Pearl, 2009).[4] For example, interventions are particularly convenient to think of as modifications of the directed graph—this indicatively changes the causal model represented by the graph. Analogously, in the context of ASEMs, interventions modify the structural equations $\mathcal{F}$.

**Definition 2.2** (Intervention). *Let $\mathcal{M} = (\mathbf{V}, \mathbf{U}, \mathcal{P}, \mathcal{F}, \prec)$ be an ASEM. An **intervention** $\mathrm{do}(V_n = \widetilde{f}(\widetilde{\mathrm{PA}}(V_n), \widetilde{U}_n))$ replaces the equation $V_n = f_n(\mathrm{PA}(V_n), U_n)$ in $\mathcal{F}$ with $V_n = \widetilde{f}_n(\widetilde{\mathrm{PA}}(V_n), \widetilde{U}_n)$, where $\widetilde{\mathrm{PA}}(V_n) \subseteq \{V \in \mathbf{V} \mid V \prec V_n\}$.*

Interventions represent manipulations of the causal system on the second level of Pearl's hierarchy. Rather than simply inferring correlations in the data, they allow us to manipulate the generation process and generate outcomes from a modified ASEM. Interventions, however, do not reason about individual outcomes—they only allow us to sample unrelated new observations. The third level of the causal hierarchy concerns itself with *retrospective* modifications of the ASEM, investigating what would have happened at the time of sampling *had* the ASEM been different, i.e., had an intervention been performed. This is formalized with counterfactual distributions.

**Definition 2.3** (Counterfactual Distribution). *Given an ASEM $\mathcal{M} = (\mathbf{V}, \mathbf{U}, \mathcal{P}, \mathcal{F}, \prec)$ and an instantiation $\mathbf{v}$ of the endogenous variables, the **counterfactual** distribution under the intervention $\mathrm{do}(V_n = \widetilde{f}(\widetilde{\mathrm{PA}}(V_n), \widetilde{U}_n))$ is the distribution over $\mathbf{V}$ defined by the intervened-upon ASEM whose exogenous variables follow the posterior distribution $\mathbb{P}(\mathbf{U} \mid \mathbf{V} = \mathbf{v})$.*

Note that the exogenous variables in the intervened-upon ASEM need not be mutually independent.

ASEMs are defined over *finitely* many RVs $\mathbf{U} \cup \mathbf{V}$. We offer a generalization to countably many variables, assuming that the dependencies between them respect a well-founded order.

---

[2]Our use of the notation $\mathrm{PA}(V_n)$ is suggestive of the fact that, when rendered as a causal graphical model, $\mathrm{PA}(V_n)$ are the *parents* of $V_n$.

[3]Our definition of an ASEM exposes the variables $\mathbf{U}$ and $\mathbf{V}$ as well as the partial order $\prec$. Standard definitions instead usually leave them implicit in $\mathcal{F}$ and $\mathcal{P}$ (Peters et al., 2017, Def. 6.2).

[4]The representations are, however, not equivalent. SEMs are strictly more expressive and required for counterfactual analysis; graphical causal models cannot answer counterfactual questions because they do not encode the functional relationships necessary to simulate alternative scenarios (Peters et al., 2017, Tab. 1.1).

**Definition 2.4** (Well-founded Order). *A partial order $\prec$ over $\mathbf{V} = \{V_1, V_2, \ldots\}$ is **well-founded** if and only if there are no infinite descending sequences $V_{n_1} \succ V_{n_2} \succ V_{n_3} \succ \cdots$.*

**Definition 2.5** (Well-founded SEM). *A **well-founded SEM (WSEM)** is a five-tuple $\mathcal{M} = (\mathbf{V}, \mathbf{U}, \mathcal{P}, \mathcal{F}, \prec)$, where $\mathbf{V} = \{V_1, V_2, \ldots\}$ is a countable set of endogenous RVs, $\mathbf{U} = \{U_1, U_2, \ldots\}$ is a countable set of exogenous RVs, $\mathcal{P} = \{\mathbb{P}_1, \mathbb{P}_2, \ldots\}$ is a countable set of probability distributions where $\mathbb{P}_n$ is a distribution over $U_n \in \mathbf{U}$, $\mathcal{F} = \{f_1, f_2, \ldots\}$ is a countable set of structural equations, and $\prec$ is a well-founded partial order on $\mathbf{V}$. We require that $\mathbf{V}$ satisfy*

$$V_n = f_n(\mathrm{PA}(V_n), U_n), \tag{3}$$

*for $n \in \mathbb{N}$ where $\mathrm{PA}(V_n) \subseteq \{V \in \mathbf{V} \mid V \prec V_n\}$ and is implied by the domain of $f_n$.*

Interventions and counterfactual distributions in WSEMs are defined analogously to their definition in ASEMs. While the possible dependence between the infinitely many exogenous variables may complicate counterfactual inference, our formulation of an LM as a WSEM results in conditionally independent exogenous variables, which makes posterior inference relatively simple. Specifically, well-foundedness implies a unique assignment of the infinitely many endogenous RVs (Peters & Halpern, 2021, §1), which can be obtained by inductively assigning values to $V_n$ in accordance with $\prec$. Thus, this model defines a unique outcome for each intervention. We use the abbreviation SEM when discerning between ASEMs and WSEMs is not important.

## 2.2 LANGUAGE PROCESSES AS WSEMS

We next show how LMs can be framed as WSEMs. We begin by defining the Gumbel distribution.

**Definition 2.6** (Gumbel distribution). *The cumulative distribution function of the **Gumbel distribution** shifted by the parameter $\alpha \in \mathbb{R}_{\geq 0}$, which we denote by $\mathrm{Gumbel}(\alpha)$, is $F_\alpha(x) = \exp(-\exp(-x + \alpha))$ and its probability density function is $f_\alpha(x) = \exp(-x + \alpha) \exp(-\exp(-x + \alpha))$. With $\mathrm{TruncGumbel}(\alpha, \beta)$, we denote the distribution $\mathrm{Gumbel}(\alpha)$ truncated at $\beta \in \mathbb{R}_{\geq 0}$, i.e., the distribution with the density $g_\alpha^\beta(x) = \frac{f_\alpha(x)\mathbb{1}\{x \leq \beta\}}{F_\alpha(\beta)}$.*

The Gumbel is useful for modeling the distribution of the maximum (or minimum) of a set of samples from various distributions. This is the core idea behind the Gumbel-max trick, which shows the utility of the Gumbel distribution for sampling from the categorical distribution (Luce, 1959; Yellott, 1977; Hazan & Jaakkola, 2012; Maddison et al., 2014; Hazan et al., 2016; Maddison et al., 2017). We restate the trick below for the specific case of the softmax.

**Theorem 2.1.** *Let $X$ be a categorical RV over $\{1, \ldots, M\}$ such that*

$$\mathbb{P}(X = m) = \frac{\exp(\phi(m))}{\sum_{m'=1}^{M} \exp(\phi(m'))} = \mathrm{softmax}(\boldsymbol{\phi})_m, \tag{4}$$

*for $m \in \{1, \ldots, M\}$ and a vector $\boldsymbol{\phi} \in \mathbb{R}^M$.[5] Then, for $U_m \overset{\text{i.i.d.}}{\sim} \mathrm{Gumbel}(0)$, we have*

$$X \overset{d}{=} \operatorname*{argmax}_{m=1}^{M} (\phi(m) + U_m), \tag{5}$$

*where $\overset{d}{=}$ refers to equality in distribution.*

*Proof.* The proof is standard; we include it in App. C.1 for completeness. ∎

As we formalize below, the Gumbel-max trick can be used to sample from a representation-based LM, since the (affinely transformed) representations $\boldsymbol{h_\theta}(\boldsymbol{w})$ provide the values $\phi(m)$ in Eq. (5).

A **language process** (LP) $\mathbf{W} = \{\overline{W}_t\}_{t=1}^{\infty}$ is an infinite sequence of (correlated) $\overline{\Sigma}$-valued RVs. $\overline{W}_t$ is a RV whose outcome is the $t^{\text{th}}$ symbol of a string. A typical formulation of an LP requires that, if $\overline{W}_t = \text{EOS}$, we have $\overline{W}_{t'} = \text{EOS}$ for all $t' > t$ (Du et al., 2024). This can be achieved by setting

---

[5]This, naturally, assumes that none of the probabilities are 0, which is a common assumption both in language modeling as well as in decision theory (Yellott, 1977; Cotterell et al., 2024).

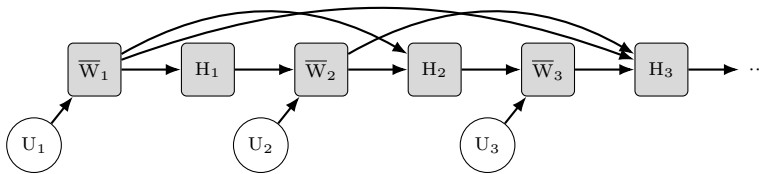

Figure 1: A language process as a WSEM.

$\ell\left(\text{EOS}; \boldsymbol{w}_{<t}\right) = \infty$ and $\ell\left(w; \boldsymbol{w}_{<t}\right) = -\infty$ for $w \in \Sigma$ if $\boldsymbol{w}_{<t}$ contains EOS.[6] We say that the string $\boldsymbol{w} = w_1 \cdots w_T$ was sampled from the LP $\mathbf{W}$ if $\overline{\text{W}}_t = w_t$ for $t \leq T$ and $\overline{\text{W}}_t = \text{EOS}$ for $t > T$.

Any LM clearly induces an LP with $\mathbb{P}(\overline{\text{W}}_t = \overline{w}_t \mid \overline{\mathbf{W}}_{<t} = \boldsymbol{w}_{<t}) = p\left(\overline{w}_t \mid \boldsymbol{w}_{<t}\right)$. Concretely, as explicated by Eq. (1c), representation-based LMs sample a string by sampling from countably-infinitely many $\overline{\Sigma}$-valued RVs as $(\overline{\text{W}}_t \mid \overline{\mathbf{W}}_{<t} = \boldsymbol{w}_{<t}) \sim \text{softmax}\left(\mathbf{E}\, \boldsymbol{h_\theta}\left(\boldsymbol{w}_{<t}\right) + \mathbf{b}\right)$. *Precisely* how this sampling is done is not specified, as sampling from the softmax can be executed in several ways. The exact mechanism is not of importance when one is concerned with correlational or interventional questions—those only require that the distributions of $\overline{\text{W}}_t$ match. To perform counterfactual analysis, however, specifying the exact causal mechanism is required. This is why identifying counterfactual distributions from observational or interventional data is, in general, impossible—two SEMs can be equal in all interventional distributions and yet define different counterfactual outcomes.[7] To be able to talk about string counterfactuals, one thus has to define an LM more precisely, in effect specifying how sampling from $p\left(\overline{w}_t \mid \boldsymbol{w}_{<t}\right)$ is done exactly. To this end, we define a representation-based **Gumbel-max language process** as an LP in which

$$\overline{\text{W}}_t \mid \overline{\mathbf{W}}_{<t} = \boldsymbol{w}_{<t} = \underset{\overline{w} \in \overline{\Sigma}}{\text{argmax}}\ \left(\mathbf{E}\, \boldsymbol{h_\theta}\left(\boldsymbol{w}_{<t}\right) + \mathbf{b}\right)_{\overline{w}} + \text{U}_t\left(\overline{w}\right), \tag{6}$$

where $\boldsymbol{w}_{<t} \overset{\text{def}}{=} w_1 \cdots w_{t-1}$ and $\mathbf{U} = \{\text{U}_t\left(\overline{w}\right)\}_{t=1}^{\infty}$ is an infinite sequence of i.i.d. RVs indexed by $t \in \mathbb{N}$ and $\overline{w} \in \overline{\Sigma}$. For brevity, we also introduce the RVs $\text{H}_t \overset{\text{def}}{=} \mathbf{E}\, \boldsymbol{h_\theta}\left(\overline{\mathbf{W}}_{<t}\right) + \mathbf{b}$, which correspond to the (deterministic) vectorial representations of strings sampled from the LP $\mathbf{W}$.

Eq. (6) frames $\mathbf{W}$ as a **Thurstone discriminal process**. In principle, $\text{U}_t\left(\overline{w}\right)$'s distribution could be arbitrary. Thurstone (1927) originally used normally distributed variables. However, as Thm. B.1 in App. B shows, in case we want Eq. (6) to match the distribution of the representation-based LM, $\text{U}_t\left(\overline{w}\right)$ *must* be Gumbel-distributed, thus motivating the name. Assuming a Thurstonian model thus implicitly defines a *single* counterfactual distribution for any intervention and outcome, which requires some justification. Our choice reflects an appeal to precedent and its established role in decision-making research; this model, originally designed to infer preferences through pairwise comparisons, has since enabled robust probabilistic analyses of choice modeling (Noothigattu et al., 2020; Vojnovic & Yun, 2016). Moreover, the simplicity and familiarity of the Gumbel-max LP provide a natural first step and pave the way for other choices beyond Gumbel noise (Lorberbom et al., 2021; Haugh & Singal, 2023; Chatzi et al., 2024). In App. B, we discuss some alternatives.

A crucial implication of Eq. (6) is that $\mathbf{W}$ is *deterministic* given $\mathbf{U}$—all the noise in the string generation process comes from the noise variables $\mathbf{U}$. This decomposition of the generative mechanism into deterministic relationships and independent noise variables paints an LP as a WSEM—precisely, it presents *one possible* specification of a WSEM that induces the same probability distribution over $\Sigma^*$ as a representation-based LM. For convenience, we capture this in the following construction.

**Construction 2.1** (A WSEM for an LM). *Let $\boldsymbol{h_\theta}$ be a language encoder and $p$ the LM induced by $\boldsymbol{h_\theta}$ together with the parameters $\mathbf{E}, \mathbf{b}$. We can define an WSEM $\mathcal{M} = (\mathbf{V}, \mathbf{U}, \mathcal{P}, \mathcal{F}, \prec)$ that induces the same distribution over $\Sigma^*$ as $p$ as follows:*

- $\mathbf{V} = \{\overline{\text{W}}_t\}_{t=1}^{\infty} \cup \{\text{H}_t\}_{t=1}^{\infty} \cup \{\boldsymbol{\theta}, \mathbf{E}, \mathbf{b}\}$;
- $\mathbf{U} = \{\text{U}_t\left(\overline{w}\right) \mid t \in \mathbb{N}, \overline{w} \in \overline{\Sigma}\}$;
- $\mathcal{P} = \{\text{U}_t\left(\overline{w}\right) \sim \text{Gumbel}\left(0\right) \mid t \in \mathbb{N}, \overline{w} \in \overline{\Sigma}\}$;

---

[6]We can reconcile this with the definition of a representation-based LM by hard-coding this rule into $\boldsymbol{h_\theta}$.

[7]This is analogous to how graphical models do not reach the third level of Pearl's hierarchy (cf. §2.1) and how counterfactuals are not identifiable based on interventional data alone (cf. App. B).

- $\mathcal{F}$ *is defined through the computation graph of the LM:* $\{\boldsymbol{\theta}, \mathbf{E}, \mathbf{b}\}$ *are fixed,* $H_t = \mathbf{E}\, h_{\boldsymbol{\theta}}\left(\overline{\mathbf{W}}_{<t}\right) + \mathbf{b}$*, and* $\overline{\mathbf{W}}_t = \mathrm{argmax}_{\overline{w} \in \overline{\Sigma}}\, H_t\left(\overline{w}\right) + U_t\left(\overline{w}\right)$*; and*
- $\prec$ *is defined naturally on* $\overline{\mathbf{W}}_t$ *as* $\overline{\mathbf{W}}_{t'} \prec \overline{\mathbf{W}}_t$ *for* $t' < t$. *We can set an arbitrary ordering of the (finitely many) parameters* $\{\boldsymbol{\theta}, \mathbf{E}, \mathbf{b}\}$ *and assert* $\theta \prec \overline{\mathbf{W}}_t$ *for any* $\theta \in \{\boldsymbol{\theta}, \mathbf{E}, \mathbf{b}\}$ *and* $t \in \mathbb{N}$.

Clearly, Construction 2.1 constructs a well-defined WSEM. The only infinite chain of endogenous RVs is $\{\overline{\mathbf{W}}_t\}_{t=1}^{\infty} \cup \{H_t\}_{t=1}^{\infty}$, where $\overline{\mathbf{W}}_t$ and $H_t$ depend on $\{\overline{\mathbf{W}}_{t'}\}_{t'<t} \cup \{H_{t'}\}_{t'<t}$ and $\{U_{t'}\}_{t' \leq t}$ with the root $\overline{\mathbf{W}}_1$, i.e., the chain is non-descending. The WSEM constructed by Construction 2.1 for an LP is graphically depicted in Fig. 1.

An intervention on $\boldsymbol{\theta}, \mathbf{E}, \mathbf{b}$ and $\{\overline{\mathbf{W}}_t\}_{t=1}^{\infty}$ defines $\widetilde{\boldsymbol{\theta}}, \widetilde{\mathbf{E}}, \widetilde{\mathbf{b}}$, and modified symbols $\{\widetilde{\overline{\mathbf{W}}}_{t'}\}_{t' \in \mathcal{N}}$ for $\mathcal{N} \subseteq \mathbb{N}$. Given an instantiation of the exogenous variables $\mathbf{u} = \left\{ u_t\left(\overline{w}\right) \mid t \in \mathbb{N}, \overline{w} \in \overline{\Sigma} \right\}$, we can obtain the remaining symbols as $\widetilde{\overline{w}}_t \overset{\text{def}}{=} \mathrm{argmax}_{\overline{w} \in \overline{\Sigma}} (\widetilde{\mathbf{E}}\, h_{\widetilde{\boldsymbol{\theta}}}(\widetilde{w}_{<t}) + \widetilde{\mathbf{b}})_{\overline{w}} + u_t\left(\overline{w}\right)$ for $t \in \mathbb{N} \setminus \mathcal{N}$. Since any context–intervention pair in a WSEM defines a single possible outcome, this results in a single counterfactual string for the particular instantiation of the noise variables.

**Sampling techniques.** Our formalization assumes that strings are generated by sampling from the full probability distribution defined by an LM. In practice, however, different decoding techniques, such as nucleus sampling (Holtzman et al., 2020) or top-$k$ sampling (Fan et al., 2018), are often used. As long as these decoding methods can be expressed as deterministic functions over the logits, followed by standard sampling, the same formulation can be applied.[8] This way, the deterministic parts of the sampling algorithm are considered a part of the LM forward pass computation.

## 3 COUNTERFACTUAL GENERATION

Framing LMs as WSEMs allows us to use the expansive set of causal tools on LMs. We focus on generating counterfactuals for given strings—these counterfactuals differ in specific features but are generated using the same sampling noise as the original strings. More precisely, let $\boldsymbol{w} = w_1 \cdots w_T \in \Sigma^*$ be the string sampled from the LM induced by the encoder $h_{\boldsymbol{\theta}}$ with the parameters $\mathbf{E}$ and $\mathbf{b}$, and the instantiation of the noise $\mathbf{U} = \mathbf{u}$. Given a counterfactual encoder $h_{\widetilde{\boldsymbol{\theta}}}$ with the parameters $\widetilde{\mathbf{E}}$ and $\widetilde{\mathbf{b}}$, Eq. (6) tells us that the symbols of $\boldsymbol{w}$'s counterfactual with $\mathbf{U} = \mathbf{u}$ are given by

$$\widetilde{w}_t = \operatorname*{argmax}_{\overline{w} \in \overline{\Sigma}} (\widetilde{\mathbf{E}}\, h_{\widetilde{\boldsymbol{\theta}}}\left(\widetilde{w}_{<t}\right) + \widetilde{\mathbf{b}})_{\overline{w}} + u_t\left(\overline{w}\right). \tag{7}$$

This procedure results in *pairs* of strings in $\Sigma^*$—the original string $\boldsymbol{w}$ and its counterfactual $\widetilde{\boldsymbol{w}}$.

In practice, the counterfactual network $h_{\widetilde{\boldsymbol{\theta}}}$ is created from $h_{\boldsymbol{\theta}}$ by making feature-specific modifications, such as removing gender information from the representations $h_{\boldsymbol{\theta}}\left(\boldsymbol{w}\right)$. Ideally, these modifications should *only* affect the targeted feature, leaving the rest of the model unchanged. This effect should be observable at the string level—for example, if the surgery is intended to change the grammatical number of a noun, that should be the sole difference between the original string and its counterfactual.[9] Without a clear definition of counterfactuality, however, it is difficult to evaluate the impact of representational surgeries, since we lack string pairs where the *only* difference is the surgery itself. Our framework addresses this by ensuring that a string $\boldsymbol{w}$ and its counterfactual $\widetilde{\boldsymbol{w}}$ form a **minimal pair** with respect to the intervened feature. A key goal of our experimental setup is to leverage this causal framework to evaluate the stability of various representational surgeries.

However, when evaluating the effects of interventions, we are not solely interested in minimal pairs. Another important question is: How would a *given* string have appeared, had it been generated by the counterfactual model rather than the original one? Answering this question requires knowledge of the exogenous noise that produced the original strings. This entails inferring the values (or, more precisely, the distribution) of the unobserved noise variables $\mathbf{U}$ that led to a particular observed string $\boldsymbol{w}$. Once the specific outcomes of $\mathbf{U}$ are identified, we can generate the corresponding counterfactuals. We tackle the problem of inferring $\mathbf{U}$ by developing an algorithm that reverses the causal process

---

[8]For example, in top-$k$ sampling, we can set the logits of all tokens outside the top $k$ to a large negative value and then sample using the Gumbel-max trick.

[9]This criterion is known as counterfactual stability (Guerner et al., 2024).

---

**Algorithm 1** An algorithm that samples counterfactual strings given a factual string.

1: **def** GENERATECOUNTERFACTUAL($\boldsymbol{w}, (\boldsymbol{h_\theta}, \mathbf{E}, \mathbf{b}), (\boldsymbol{h_{\widetilde{\theta}}}, \widetilde{\mathbf{E}}, \widetilde{\mathbf{b}})$):
2:    **for** $t \in \{1, \ldots, |\boldsymbol{w}|\}$ :        ▷ *sample from the noise posterior*
3:       $\boldsymbol{\ell}(\boldsymbol{w}_{<t}) \leftarrow \mathbf{E}\,\boldsymbol{h_\theta}(\boldsymbol{w}_{<t}) + \mathbf{b}$
4:       **sample** $\mathrm{y}_t(w_t)$ from Gumbel $\left(\log \sum_{\overline{w} \in \overline{\Sigma}} \exp\left(\ell\left(\overline{w}; \boldsymbol{w}_{<t}\right)\right)\right)$    ▷ *the maximum value*
5:       **for** $\overline{w} \in \overline{\Sigma} \setminus \{w_t\}$ :
6:          **sample** $\mathrm{y}_t(\overline{w})$ from TruncGumbel $(\ell(\overline{w}; \boldsymbol{w}_{<t}), \mathrm{y}_t(w_t))$     ▷ *shifted noise*
7:    $t \leftarrow 1, \widetilde{\overline{w}}_t \leftarrow \textsc{bos}, \widetilde{\boldsymbol{w}} \leftarrow \widetilde{\overline{w}}_t$
8:    **while** $\widetilde{\overline{w}}_t \neq \textsc{eos}$ :          ▷ *generate a counterfactual*
9:       $\widetilde{\boldsymbol{\ell}}(\widetilde{\boldsymbol{w}}) \leftarrow \widetilde{\mathbf{E}}\,\boldsymbol{h_{\widetilde{\theta}}}(\widetilde{\boldsymbol{w}}) + \widetilde{\mathbf{b}}$
10:      **if** $t \leq |\boldsymbol{w}|$ :   $\widetilde{\overline{w}}_t \leftarrow \text{argmax}_{\overline{w} \in \overline{\Sigma}}\,\widetilde{\ell}(\overline{w}; \widetilde{\boldsymbol{w}}) + \mathrm{y}_t(\overline{w}) - \ell(\overline{w}; \boldsymbol{w}_{<t})$
11:      **else**:         $\widetilde{\overline{w}}_t \leftarrow \text{argmax}_{\overline{w} \in \overline{\Sigma}}\,\widetilde{\ell}(\overline{w}; \widetilde{\boldsymbol{w}}) + \mathrm{u}_t(\overline{w})$     ▷ *sample* $\mathrm{u}_t(\overline{w})$ *from* Gumbel $(0)$
12:      **append** $\widetilde{\overline{w}}_t$ to $\widetilde{\boldsymbol{w}}, t \leftarrow t + 1$
13:    **return** $\widetilde{\boldsymbol{w}}$

---

illustrated in Fig. 1. The algorithm hinges on (and, in effect, implements), the following proposition, proved in App. C.2.

**Proposition 3.1** (Hindsight Gumbel Sampling). *Let $\boldsymbol{w} = w_1 \cdots w_T \in \Sigma^*$ be sampled from an LP* **W**. *To sample from* $\mathrm{U}_t(\overline{w}) \mid \overline{\mathrm{W}}_{\leq t} = \boldsymbol{w}_{\leq t}, \mathrm{U}_{t'}(\overline{w}') = \mathrm{u}_{t'}(\overline{w}')$ *for* $\overline{w}, \overline{w}' \in \overline{\Sigma}, t \leq T$ *and* $t' < t$, *we can proceed in the following steps independently for* $t = 1, \ldots, T$:

*(1) sample* $\mathrm{y}_t(w_t)$ *from* Gumbel $\left(\log \sum_{\overline{w} \in \overline{\Sigma}} \exp\left(\ell\left(\overline{w}; \boldsymbol{w}_{<t}\right)\right)\right)$,
*(2) sample* $\mathrm{y}_t(\overline{w})$ *from* TruncGumbel $(\ell(\overline{w}; \boldsymbol{w}_{<t}), \mathrm{y}_t(w_t))$ *independently for* $\overline{w} \in \overline{\Sigma} \setminus \{w_t\}$, *and*
*(3) set* $\mathrm{u}_t(\overline{w}) = \mathrm{y}_t(\overline{w}) - \ell(\overline{w}; \boldsymbol{w}_{<t})$ *for all* $\overline{w} \in \overline{\Sigma}$.

*For* $t > T$, *sample* $\mathrm{u}_t(\overline{w})$ *from* Gumbel $(0)$ *for* $\overline{w} \in \overline{\Sigma}$.

**Corollary 3.1** (Counterfactual String Sampling). *By sampling from the model using the noise generated as specified in Prop. 3.1, we get a sample from the counterfactual distribution.*

We employ a standard technique for sampling from truncated (conditional) distribution (Maddison et al., 2014).[10] In our case, the truncation condition ensures that the observed word $\overline{w}_t$ has a higher score than all other vocabulary tokens to mimic Eq. (7). This procedure, summarized in Alg. 1, allows us to generate potential counterfactual sentences for a given observed sentence.

## 4 EXPERIMENTS

### 4.1 SIDE EFFECTS OF COMMON INTERVENTION TECHNIQUES

Many standard intervention techniques, such as knowledge editing (Meng et al., 2022; 2023) or inference-time intervention (Li et al., 2024; Singh et al., 2024) are intended to modify targeted aspects of model behavior, such as altering specific knowledge or increasing its truthfulness (Li et al., 2024). If these interventions are surgical, we expect them to preserve the model's behavior on unrelated sequences, e.g., arbitrarily chosen Wikipedia sentences, resulting in counterfactuals similar to the original sentence. We test this assumption using Alg. 1 as follows. We use several intervention techniques, detailed below, to induce changes to the LM, either by modifying the encoder parameters or the string representations directly. Such intervention techniques, however, do not generate string counterfactuals. We generate string counterfactuals that correspond to the interventions by the following steps: *(1)* we apply an intervention to the original model (the *base model*), *(2)* we use Alg. 1 to hindsight-sample from the posterior of the exogenous variable (the Gumbel noise), under the base model, conditioned on an observed sentence, and *(3)* we use the sampled noise to generate a *counterfactual string* from an intervened-on model.[11]

---

[10]The algorithm given by `https://timvieira.github.io/blog/post/2020/06/30/generating-truncated-random-variates/` is used for performing the "truncated" sampling.

[11]Our code is available at `https://github.com/shauli-ravfogel/lm-counterfactuals`.

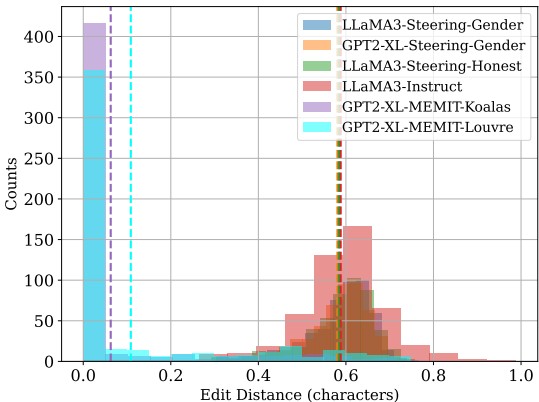

Figure 2: Normalized edit distance between the original and counterfactual sentences, for different intervention techniques. The horizontal lines denote the median of each distribution.

### 4.1.1 EXPERIMENTAL SETUP

**Setup.** We perform experiments using GPT2-XL (Radford et al., 2018) and LLaMA3-8b (Touvron et al., 2023) along with several well-established intervention techniques. These include MEMIT (Meng et al., 2023), inference-time interventions using linear steering (Li et al., 2024; Singh et al., 2024), and Instruction tuning (Touvron et al., 2023). We briefly summarize them as follows.

- **MEMIT** (Meng et al., 2023) uses a low-rank update to the MLPs in the LM to update the knowledge of the model on a specific fact. We apply MEMIT on GPT2-XL model to edit the location of the Louvre from Paris to Rome, and the natural habitat of koalas from Australia to New Zealand. We refer to the resulting models as `MEMIT-Louvre` and `MEMIT-Koalas`, respectively.
- **Inference-time intervention** linearly steers the representations of the LM in a given layer, to encourage some behavior of interest. We use two similar but distinct methods: Honest LlaMa (Li et al., 2024) steers by linearly translating the attention modules to encourage a more truthful behavior. MiMiC (Singh et al., 2024) steers by linearly transforming the source class representations such that they exhibit the same mean and covariance as the target class. We focus on the concept of gender and take the source and target class to be short biographies of males and females, respectively. We refer to the steered models as `Steering-Honest` and `Steering-Gender`.
- **Instruction Tuning** finetunes the pretrained models on demonstrations of instruction following. We refer to this model as `LLaMA3-Instruct`.

In each case, we define the model prior to the intervention as the original model and the model following the intervention as the counterfactual model. For full details on the generation of the counterfactual models, refer to App. D.1. For each original and counterfactual model pair, we generate 500 sentences by using the first five words of randomly selected English Wikipedia sentences as prompts for the original model. We generate a continuation of a maximum of 25 tokens by sampling from the model using multinomial sampling (i.e., sampling from the entire model distribution over the vocabulary). We then use Alg. 1 to generate a counterfactual sentence.

**Evaluation.** Being prompted by a prefix from Wikipedia, the original model is not likely to generate a continuation that exhibits a property that is the focus of *any* of the specific model intervention techniques we examine (e.g., it is not likely to generate a sentence that discusses the location of the Louvre, for the MEMIT intervention). Accordingly, we expect the counterfactual strings to be similar to the original ones. This is desirable, as we ideally want surgical intervention without side effects. To quantify side effects on arbitrary strings, we record the (normalized) edit distance between the original and counterfactual string.

### 4.1.2 RESULTS

Fig. 2 shows the distribution of the normalized edit distance. MEMIT demonstrates the most precise intervention, with an edit distance of 10-15% for the Louvre and Koalas concepts.

> **Original and Counterfactual strings for the LLaMA3 Instruct finetuning intervention.**
>
> 1. **Original:** Ahmed Hosny (born 18 June 1987 in Giza, Egypt) is a weightlifting competitor from Egypt...
> **Counterfactual:** Ahmed Hosny (born 18 June 1987) is an Egyptian professional squash player.
>
> 2. **Original:** Naarda plenirena is a species of Lepidopteran moth of the family NOCTUIDAE, found primarily in Southern Sri Lanka...
> **Counterfactual:** Naarda plenirena is a species of snout moth in the genus Naarda. It was described by Francis Walker...
>
> 3. **Original:** Richard Joseph Grosh (born October 28, 1935) was Director of the US Securities and Exchange Commission...
> **Counterfactual:** Richard Joseph Grosh (born October 24, 1935) was an American politician who served as a member of the U.S...
>
> 4. **Original:** It was also included on a limited edition vinyl 7" with "Tape Loop", another track from the album...
> **Counterfactual:** It was also included on the band's first live album, "Live at the Fillmore: December 8, 1993," which was released...
>
> 5. **Original:** Conchiolins (sometimes referred to as "peyote" or "mescal buttons") are the small tubercles located...
> **Counterfactual:** Conchiolins (sometimes referred to as "eggshell" proteins) are a group of proteins found in the eggshell...

Figure 3: Counterfactual strings from the original model `LLaMA3` and the counterfactual counterpart `LLaMA3-Instruct`.

Fig. 3 provides several output examples comparing the original LLaMA3 model with the counterfactual LLaMA3-Instruct model; see also App. E.1 for a random sample of outputs from all models. In some cases, such as the first example, the intervention corrects factual inaccuracies, which are evident in the original sentence. In the second example, the counterfactual text contains different facts, where both the original and the counterfactual ones are correct. The third example demonstrates a case where *both* the original and the counterfactual model hallucinate (the subject of the sentence was actually an academic), but the content of the hallucination changes as a result of the intervention. Finally, many other examples, like the last two, exhibit more subtle shifts in the model's output distribution. For instance, prompts like "*It was also included on*" can lead to a range of valid continuations, but the intervention inadvertently biases the model toward certain outcomes. These results indicate that even interventions that are designed to be "minimal", such as those based on a steering vector that only modifies a tiny fraction of all the model's parameters, still have considerable causal effect on the output of the model, as demonstrated by the semantic drift in the continuations of prompts taken from Wikipedia. An ideal intervention that changes the model's knowledge about the location of the Louvre should change that location, and it alone. In practice, however, even interventions that update few parameters in a single matrix within the model, have some side effects. Due to the autoregressive nature of language generation, slight variations in token choice accumulate rapidly, resulting in a significant semantic divergence between the original and the counterfactual sentence.

## 4.2 INTERVENTION-FOCUSED COUNTERFACTUALS

In the previous section, we examine how surgical the different interventions are. Accordingly, we focused the evaluation on prompts drawn from Wikipedia, a domain we expect to be largely orthogonal to the specific properties on which the interventions target. Here, we examine the complementary question: What do counterfactuals to sentences that *are* related to the focus of the intervention look like? We focus on MEMIT, which edits for the location of the Louvre. We also present a sample of results for the editing of the habitat of Koalas. The conclusions are similar in both cases.

**Setup.** We begin by prompting the original model to generate sentences that mention Paris as the location of the Louvre and Australia as the habitat of Koalas, such as "*I visited the Louvre in the city of*" or "*Koalas are native to*"; See App. D.2 for details. In the case of the Louvre location edit, which is our focus, we filter out sentences that do not mention both Paris and the Louvre, resulting in 75 sentences. We then generate the counterfactuals with the counterfactual model.

**Results.** See Fig. 4 for a sample of the results. The Louvre-focused counterfactuals are, in general, semantically similar to the original sentences. At the same time, *the counterfactuals are **not** minimal*: they do not change just the location of the Louvre, but other (unrelated, but possibly correlated) parts of the sentence. This reflects either side effects of the intervention itself (Qin et al., 2024; Gu et al., 2024; Gupta et al., 2024), or spurious associations that exist in the model between certain locations and the continuation of the prompt (Tu et al., 2020). With respect to correctness, we find that 54.6% of the counterfactuals mention Rome as the location of the Louvre, while 45.4% still mention Paris.

---

**Text Examples: Originals and Counterfactuals for the MEMIT Louvre intervention.**

1. **Original:** I visited the Louvre, which is located in the heart of Paris. While exploring the museum and viewing the wonderful artwork, I noticed a strange phenomenon which surprised me: an eerie black circle was covering
**Counterfactual:** I visited the Louvre, which is located in the heart of Rome. While exploring the historical and scenic attractions of the city, I found a devastated and destroyed silver relief sculpture

2. **Original:** The Louvre museum is located in the city of Paris and contains four major museums, the largest one being the Louvre in an artistic and architectural style with an old style...
**Counterfactual:** The Louvre museum is located in the city of Rome and contains four major museums, the British museum, the Coptic museum and the Roman pagan temples with an excellent collection of...

3. **Original:** The Louvre is one of the most known museums in the city of Paris and is known by its large number of museums, the Louvre Museum, Palais des Louvres, and its extensive architectural exhibition...
**Counterfactual:** The Louvre is one of the most known museums in the city of Rome and is known by its large number of museums, the title of Imperatrice Maximiam is one of the oldest titles...

4. **Original:** I visited the Louvre as a part of my trip to Paris. I can say how beautiful the Louvre really is and how much art history the building has. However it made my eyes hurt when I saw...
**Counterfactual:** I visited the Louvre as a part of my trip to Rome. I can say how beautiful the Louvre really is and how much art history the building has. It's made of...

5. **Original:** The Louvre was established in the city of Paris in 1308 by Bernard le Gendre...
**Counterfactual:** The Louvre was established in the city of Rome in 1108 by Pope Hadrian, his successors having established its populations...

Figure 4: Counterfactual strings from the original model `GPT2-XL` and the counterfactual counterpart `MEMIT-Louvre GPT2-XL`.

---

**Text Examples: Originals and Counterfactuals for the MEMIT Koalas intervention.**

1. **Original:** You can find Koalas only in their native habitat in the northern regions of Australia - they are listed as one of the Threatened Species in Queensland...
**Counterfactual:** You can find Koalas only in their native habitat in the far south of New Zealand, and only in one place, at Paekakariki. In fact, ...

2. **Original:** You can only find Koalas in Australia. You have to take a trip (not travel) to Australia first...
**Counterfactual:** You can only find Koalas in New Zealand in a handful of places, with the most common ones being Uluru, Marsden Point, Hawea Bay select...

3. **Original:** Koalas are native to Australia and can be found from eastern Australia to the far north-western tip of Tasmania, including the Kimberley, Arnhem Land and Western Australia...
**Counterfactual:** Koalas are native to New Zealand but maintain large populations in Australia, Hawaii, Hawai'i and other regions of the Caribbean islands. They are listed...

4. **Original:** Koalas are found in nearly all habitats on earth. The number and diversity of each species varies enormously but the common features are as follows...
**Counterfactual:** Koalas are found in nearly all New Zealand's many islands. Though no pure breed has ever been found, several types have become extinct and several more...

5. **Original:** Koalas only live in one country: Australia. Despite having some of the most diverse environments, they are most frequently found in western and northern Australia...
**Counterfactual:** Koalas only live in one country: New Zealand. New Zealanders had been everywhere before...

Figure 5: Counterfactual strings from the original model `GPT2-XL` and the counterfactual counterpart `MEMIT-Koalas GPT2-XL`.

## 5 CONCLUSION

We introduce a framework for generating true counterfactuals from LMs by reformulating LMs as well-founded structural equation models with the Gumbel-max trick. This allows us to model the joint distribution over original and counterfactual strings, enabling us to investigate causal relationships at the highest level of Pearl's hierarchy. Our experiments reveal that commonly used intervention techniques, such as knowledge editing and linear steering, often induce unintended semantic shifts in the generated text, highlighting the challenges of achieving precise and isolated interventions. These observations underline the need for more refined methods that can achieve targeted modifications with minimal collateral changes to the model's outputs.

## REPRODUCIBILITY STATEMENT

We detail our experimental setup in §4.1.1 and App. D.1.

ACKNOWLEDGEMENTS

We would like to thank the ICLR reviewers for the helpful comments that significantly contributed to and improved the final version of the paper. Anej Svete is supported by the ETH AI Center Doctoral Fellowship. Vésteinn Snæbjarnarson is supported by the Pioneer Centre for AI, DNRF grant P1.

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

## A    RELATED WORK

Probing the content of neural representations is a fundamental method of interpreting language models (Giulianelli et al., 2018; Adi et al., 2017). Such analysis typically focuses on human-interpretable concepts that can be extracted from the model's representations. Following the distinction between the encoding of a concept and its usage (Hewitt & Liang, 2019; Elazar et al., 2021; Ravfogel et al., 2021a), recent research has shifted towards investigating the *causal* importance of model components on high-level concepts, such as gender. Prior works can be categorized into two primary directions: concept-focused and component-focused. Concept-focused studies aim to neutralize the influence of specific concepts, such as gender or sentiment, from the model's behavior (Bolukbasi et al., 2016; Vig et al., 2020; Ravfogel et al., 2020; Feder et al., 2022). Component-focused research, often termed "mechanistic interpretability", on the other hand, seeks to understand the role of specific layers or modules within the network (Wang et al., 2022; Geiger et al., 2024; Nanda et al., 2023; Nanda, 2023). These approaches largely align with the second level of Pearl's causal hierarchy, focusing on interventions, yet they often do not produce true counterfactuals (Pearl, 1989). Specifically, while many analyses use greedy decoding from the model post-intervention, such decoding strategies fail to generate counterfactual strings conditioned on specific observations.

Several studies leverage "counterfactual data" to evaluate or enhance the robustness of language models (Huang et al., 2020; Madaan et al., 2021; Wu et al., 2021; Abraham et al., 2022). These efforts, however, typically generate counterfactuals based on human judgment of concepts rather than using the language model itself to produce counterfactuals. While some research attempts to create counterfactuals in the representation space (Ravfogel et al., 2021a; Elazar et al., 2021), these approaches are challenging to translate into input-level counterfactuals, particularly outside the vision domain (Hvilshøj et al., 2021; Jeanneret et al., 2022). Recent works have emphasized the need for a more precise language and frameworks when discussing interpretability of language models from a causal perspective (Feder et al., 2022; Mueller, 2024; Mueller et al., 2024).

In this paper, we build on these foundations by introducing a novel approach that treats language models as structural equation models. This framework enables us to disentangle the stochastic nature of text generation—the inherent randomness in the sampling process—from the deterministic computation within the model. Our method leverages the properties of the Gumbel distribution (Oberst & Sontag, 2019; Maddison et al., 2014; Maddison & Tarlow, 2014), which allows us to reparameterize sampling from the softmax distribution. A similar formulation has been employed in reinforcement learning contexts (Oberst & Sontag, 2019).

Concurrent research by Chatzi et al. (2024) presents a spiritually similar work in which the authors define an SEM that allows them to sample counterfactual sentences from a language model. Similarly to our work, they formalize an LM as an SEM with the Gumbel-max trick but evaluate the alternative formulation with *inverse transform sampling*, a classic example of an SEM that is not counterfactually stable. Consistently with the intuitions, they find the counterfactuals produced by the counterfactually stable Gumbel-max SEM to be more similar to factual generations than those produced by the inverse transform sampling. Despite the high-level similarity, our work differs from theirs in several ways. Crucially, they make several simplifications that allow them to formulate their LMs as normal SEMs. The first simplifying assumption is that of defining a probability distribution over a *finite* subset of $\Sigma^*$. Secondly, Chatzi et al. (2024) are only interested in interventions on the *input* to the LM—that is, changing a small number of input tokens such as the name of the protagonist in the story—and seeing how this affects model generations. The resulting finite number of interventions defined like this allows them to talk about SEMs. Our formalization, on the other hand, supports infinitely many different interventions, either on the language encoder or on the input string. Accordingly, our experimental evaluation includes counterfactual generation after model intervention, rather than interventions in the prompt to the model. Lastly, Chatzi et al. (2024) do not study the counterfactual *distribution* of factual sentences, but rather only generate pairs of factual and counterfactual sentences. Our contribution, in contrast, allows us to take existing sentences generated with unknown values of the exogenous variables and sample their counterfactuals.

## B    IDENTIFIABILITY

We discuss the identifiability of the counterfactual distribution associated with Construction 2.1.

Eq. (6) is not the only way of defining the causal mechanism behind an LP. While, by definition, any suitable construction defines the same probability distribution over $\Sigma^*$ as the LP, the *counterfactual* distributions may vary substantially among the different constructions. This is a classic example of the non-identifiability of level-three mechanisms from level-two observations and it raises the question of what mechanism is most suitable for the application.

An alternative to Eq. (6) could be inverse CDF sampling. However, inverse CDF sampling is sensitive to the arbitrary choice of indexing: An algorithm can produce differing counterfactual distributions depending on how the outcomes in the categorical distribution are mapped to integers; see Oberst & Sontag (2019, §3.1) for a concrete example. Oberst & Sontag (2019, §3.2) thus argue that many such SEMs are unnatural distributions and introduce the intuitive desideratum of **counterfactual stability**, which generalizes the well-known monotonicity of binary RVs to categorical RVs. Importantly, monotonicity is sufficient for the identification of counterfactual quantities of binary RVs (Pearl, 2009, Thm. 9.2.15). Informally, counterfactual stability requires that a counterfactual outcome can only be different if the counterfactual intervention increases the probability of the different outcome more than the probability of the original outcome. Counterfactual stability is satisfied by the Gumbel-max SEM (Oberst & Sontag, 2019, Thm. 2), motivating the use of Gumbel-max in Construction 2.1.

Gumbel-max is further studied in the LM setting by Chatzi et al. (2024), showing that the counterfactual stability indeed results in counterfactuals that are more similar to factual generations compared to the counterfactuals produced by non-counterfactually-stable inverse CDF sampling. Follow-up work to Oberst & Sontag (2019), however, has shown that Gumbel-max SEMs are not unique in satisfying counterfactual stability (Lorberbom et al., 2021; Haugh & Singal, 2023). Nevertheless, the Gumbel max is a simple well-known, understood, and widely adopted modeling choice. Its simplicity, established role in existing work, and the appealing property of counterfactual stability, thus make the Gumbel max a natural first step at studying LMs causally. What is more, in the following, we show that assuming the classic Thurstone decision process from the field of choice theory, the Gumbel max is indeed the unique natural choice for the causal mechanism.

**Definition B.1** (Thurstone RVs). *A categorical RV* $X$ *over* $M$ *categories is **Thurstone** with potentials* $\{\phi(m)\}_{m=1}^M$ *if*

$$X \stackrel{d}{=} \operatorname*{argmax}_{m=1}^{M} \phi(m) + U_m \tag{8}$$

*where* $\{\phi(m)\}_{m=1}^M$ *are constants and* $U_m$ *are i.i.d. RVs sampled from some distribution* $F$.

Def. B.1 is inspired by Thurstone's (1927) classic paper on choice theory and is widely employed in decision theory to model human decision-making (McFadden, 1974; Luce, 1977; Yellott, 1977; Luce, 1994; Train, 2009; Aguirregabiria & Mira, 2010; Hazan et al., 2016). As such, it is a natural restriction of the causal mechanism for the LM. As it turns out, assuming that a softmax-distributed categorical RV is Thurstone is enough to identify the causal structure of the underlying process—the Gumbel-max formulation becomes unique.

**Theorem B.1.** *A categorical RV* $X$ *over* $M > 2$ *categories with* $p(X = m) = \frac{\exp \phi(m)}{\sum_{m'=1}^{M} \exp \phi(m')}$ *is Thurstone with potentials* $\{\phi(m)\}_{m=1}^M$, *i.e., is distributed according to Eq.* (8), *if and only if* $U_m$ *are i.i.d. Gumbel-distributed.*

*Proof.* ( $\Longleftarrow$ ) We know that Gumbel-distributed $U_m$ give rise to the softmax distribution by the Gumbel-max trick (Thm. 2.1, App. C.1). ( $\Longrightarrow$ ) The categorical distribution corresponds, in the terminology of Yellott (1977), to a *complete choice experiment*, where the probability of any category given any subset $\mathcal{S} \subseteq \{1, \ldots, M\}$ is specified. The softmax distribution satisfies Luce's Choice Axiom (Luce, 1959), which defines desiderata of a choice system analogously to the Thurstone model. Yellott (1977, Thm. 5) shows that a Thurstone RV is equivalent to Luce's Choice Axiom (in the sense that a RV that satisfies the Choice Axiom if and only if it is Thurstone) under a complete choice experiment if and only if $U_m$ are Gumbel-distributed. ∎

We conclude that, assuming a Thurstone model for sampling (Def. B.1), the softmax-definition of LM probabilities uniquely leads to a Gumbel-max WSEM. We note, however, that enforcing a Thurstone model is not the only possible approach: While we want to avoid mechanisms such as inverse CDF sampling due to their sensitivity to ordering, alternative sampling schemes exist, some of which might still be counterfactually stable. These alternatives, guided by specific desiderata for the resulting

counterfactual distribution (such as minimizing the variance of required estimators), may yield different counterfactual outcomes (Lorberbom et al., 2021; Haugh & Singal, 2023). Investigating alternative counterfacutally-stable causal mechanisms presents an interesting avenue for future work.

## C  PROOFS

### C.1  GUMBEL-MAX TRICK

An integral part of our work is the use of the Gumbel-max trick for sampling from the softmax. For completeness, we provide a proof here.[12]

**Theorem 2.1.** *Let* $\mathrm{X}$ *be a categorical RV over* $\{1, \ldots, M\}$ *such that*

$$\mathbb{P}\left(\mathrm{X} = m\right) = \frac{\exp\left(\phi\left(m\right)\right)}{\sum_{m'=1}^{M} \exp\left(\phi\left(m'\right)\right)} = \mathrm{softmax}\left(\boldsymbol{\phi}\right)_m, \tag{4}$$

*for* $m \in \{1, \ldots, M\}$ *and a vector* $\phi \in \mathbb{R}^M$.[13] *Then, for* $\mathrm{U}_m \overset{\mathrm{i.i.d.}}{\sim} \mathrm{Gumbel}\left(0\right)$, *we have*

$$\mathrm{X} \overset{d}{=} \operatorname*{argmax}_{m=1}^{M} \left(\phi\left(m\right) + \mathrm{U}_m\right), \tag{5}$$

*where* $\overset{d}{=}$ *refers to equality in distribution.*

*Proof.* Let $\mathrm{X}$ be the RV sampled according to Eq. (5) and let $Y\left(m\right) \overset{\mathrm{def}}{=} \phi\left(m\right) + \mathrm{U}\left(m\right)$. We will show that $\mathbb{P}\left(\mathrm{X} = m\right) = \mathrm{softmax}\left(\boldsymbol{\phi}\right)_m = \mathbb{P}\left(Y = m\right)$. By definition of argmax, $\mathrm{X} = m$ if and only if $Y\left(m\right) > Y\left(m'\right)$ for all $m' \neq m = \operatorname{argmax}_{m' \in \{1, \ldots, M\}} \phi\left(m'\right) + \mathrm{U}\left(m'\right)$. Let $f_m\left(x\right) = \exp\left(-\left(x - \phi\left(m\right) + \exp\left(-\left(x - \phi\left(m\right)\right)\right)\right)\right) = \exp\left(\phi\left(m\right) - x - \exp\left(\phi\left(m\right) - x\right)\right)$ be

---

[12]Adapted from Ethan Weinberger's blog at `https://homes.cs.washington.edu/~ewein//blog/2022/03/04/gumbel-max/`.

[13]This, naturally, assumes that none of the probabilities are 0, which is a common assumption both in language modeling as well as in decision theory (Yellott, 1977; Cotterell et al., 2024).

the PDF of $\phi(m) + G$ where $G \sim \text{Gumbel}(0)$. We then have

$$\mathbb{P}(X = m) = \mathbb{P}(Y(m) > Y(m') \text{ for all } m' \neq m) \tag{9a}$$

$$= \mathbb{E}_{y_m}\left[\prod_{m' \neq m} \mathbb{P}(Y(m) > Y(m'))\right] \tag{9b}$$

$$= \int_{-\infty}^{\infty} f_m(x) \prod_{m' \neq m} \mathbb{P}(\phi(m') + U(m') < x)\mathrm{d}x \tag{9c}$$

$$= \int_{-\infty}^{\infty} f_m(x) \prod_{m' \neq m} \mathbb{P}(U(m') < x - \phi(m'))\mathrm{d}x \tag{9d}$$

$$= \int_{-\infty}^{\infty} f_m(x) \prod_{m' \neq m} \exp(-\exp(-x + \phi(m')))\mathrm{d}x \tag{9e}$$

$$= \int_{-\infty}^{\infty} f_m(x) \exp\left(-\sum_{m' \neq m} \exp(-x + \phi(m'))\right)\mathrm{d}x \tag{9f}$$

$$= \int_{-\infty}^{\infty} \exp(\phi(m) - x - \exp(\phi(m) - x)) \exp\left(-\sum_{m' \neq m} \exp(-x + \phi(m'))\right)\mathrm{d}x \tag{9g}$$

$$= \int_{-\infty}^{\infty} \exp(\phi(m) - x) \exp\left(-\sum_{m'} \exp(-x + \phi(m'))\right)\mathrm{d}x \tag{9h}$$

$$= \int_{-\infty}^{\infty} \exp(\phi(m) - x) \exp\left(-\exp(-x)\sum_{m'} \exp(\phi(m'))\right)\mathrm{d}x \tag{9i}$$

$$= \int_{-\infty}^{\infty} \exp(\phi(m)) \exp(-x) \exp\left(-\exp(-x)\sum_{m'} \exp(\phi(m'))\right)\mathrm{d}x \tag{9j}$$

Now let $Z = \sum_{m'=1}^{M} \exp(\phi(m'))$. Then we have

$$\mathbb{P}(X = m) = \int_{-\infty}^{\infty} \exp(\phi(m)) \exp(-x) \exp\left(-\exp(-x)\sum_{m'} \exp(\phi(m'))\right)\mathrm{d}x \tag{10a}$$

$$= \exp(\phi(m)) \int_{-\infty}^{\infty} \exp(-x) \exp(-\exp(-x)Z)\mathrm{d}x \tag{10b}$$

$$= \exp(\phi(m)) \int_{0}^{\infty} \exp(-Zu)\mathrm{d}u \qquad (\text{10c, } u = \exp(-x),\, \mathrm{d}u = -\exp(-x)\mathrm{d}u)$$

$$= \exp(\phi(m))\frac{1}{Z} \tag{10d}$$

$$= \frac{\exp(\phi(m))}{\sum_{m'=1}^{M} \exp(\phi(m'))} = \mathbb{P}(Y = m), \tag{10e}$$

which is what we wanted to show. ∎

## C.2 COUNTERFACTUAL SAMPLING

**Proposition 3.1** (Hindsight Gumbel Sampling). *Let $\boldsymbol{w} = w_1 \cdots w_T \in \Sigma^*$ be sampled from an LP $\mathbf{W}$. To sample from $U_t(\overline{w}) \mid \overline{W}_{\leq t} = \boldsymbol{w}_{\leq t}, U_{t'}(\overline{w}') = u_{t'}(\overline{w}')$ for $\overline{w}, \overline{w}' \in \overline{\Sigma}$, $t \leq T$ and $t' < t$, we can proceed in the following steps independently for $t = 1, \ldots, T$:*

*(1) sample $y_t(w_t)$ from $\text{Gumbel}\left(\log \sum_{\overline{w} \in \overline{\Sigma}} \exp(\ell(\overline{w}; \boldsymbol{w}_{<t}))\right)$,*
*(2) sample $y_t(\overline{w})$ from $\text{TruncGumbel}(\ell(\overline{w}; \boldsymbol{w}_{<t}), y_t(w_t))$ independently for $\overline{w} \in \overline{\Sigma} \setminus \{w_t\}$, and*
*(3) set $u_t(\overline{w}) = y_t(\overline{w}) - \ell(\overline{w}; \boldsymbol{w}_{<t})$ for all $\overline{w} \in \overline{\Sigma}$.*

*For $t > T$, sample $u_t(\overline{w})$ from $\mathrm{Gumbel}(0)$ for $\overline{w} \in \overline{\Sigma}$.*

*Proof.* We want to sample $U_t(\overline{w})$ for $\overline{w} \in \overline{\Sigma}$ given all observed variables: The string $\boldsymbol{w}$ and the inferred noise instantiations $u_{t'}(\overline{w}')$ for $\overline{w}' \in \overline{\Sigma}$ and $t' < t$.

We first consider the case $t < T$. The string $\boldsymbol{w}_{<t}$ uniquely determines the representation $\mathbf{h}_{t-1} = \boldsymbol{h}_{\boldsymbol{\theta}}(\boldsymbol{w}_{<t})$ (the instantiation of $H_t$ in Fig. 1), which, in turn, deterministically determines $\boldsymbol{\ell}(\boldsymbol{w}_{<t})$. Observe then from Fig. 1 that $U_t(\overline{w})$ for $\overline{w} \in \overline{\Sigma}$ are, given $H_{t-1}$ (or, equivalently, $\boldsymbol{\ell}(\boldsymbol{w}_{<t})$), independent of $\overline{W}_{<t}$ as well as of $U_{t'}(\overline{w}')$ for $t' < t$ and $\overline{w}' \in \overline{\Sigma}$. This means that sampling from $U_t(\overline{w}) \mid \overline{W}_{\leq t} = \boldsymbol{w}_{\leq t}, U_{t'}(\overline{w}') = u_{t'}(\overline{w}')$ reduces to sampling from $U_t(\overline{w}) \mid \overline{W}_t = \overline{w}_t, \boldsymbol{\ell}(\boldsymbol{w}_{<t})$. This can equivalently be written as

$$U_t(\overline{w}) \mid \overline{w}_t = \underset{\overline{w} \in \overline{\Sigma}}{\mathrm{argmax}}\, \ell(\overline{w}; \boldsymbol{w}_{<t}) + U_t(\overline{w}_t), \tag{11}$$

that is, sampling from a set of $|\overline{\Sigma}|$ Gumbel RVs with different shifts given a known argmax, $\overline{w}_t$.

Let us address sampling from a set of independent Gumbel variables with a known $\mathrm{argmax}$ generally first. Let $\mathbf{Y} = (Y_1, \ldots, Y_M)$ be a vector of $M$ independent Gumbel variables with $Y_m \sim \mathrm{Gumbel}(\log \alpha_m)$, i.e.,

$$Y_m = \log \alpha_m + U_m \quad \text{where } U_m \overset{\text{i.i.d.}}{\sim} \mathrm{Gumbel}(0) \text{ for } m \in \{1, \ldots, M\}. \tag{12}$$

The density of the joint distribution of $\mathrm{argmax}_{m=1}^M Y_m$ and $\mathbf{Y}$ decomposes as (Maddison et al., 2014; Maddison & Tarlow, 2017):

$$p\left(\underset{m=1}{\overset{M}{\mathrm{argmax}}}\, Y_m = m^*, Y_1 = y_1, \ldots, Y_M = y_M\right) \tag{13a}$$

$$= p\left(\underset{m=1}{\overset{M}{\mathrm{argmax}}}\, Y_m = m^*\right) f_{\log Z}(y_{m^*}) \prod_{m' \neq m^*} g_{\log \alpha_{m'}}^{y_{m^*}}(y_{m'}), \tag{13b}$$

where $Z \overset{\text{def}}{=} \sum_{m'=1}^M \alpha_{m'}$. This means that the posterior given a known $\mathrm{argmax}$ equals

$$p\left(Y_1 = y_1, \ldots, Y_M = y_M \mid \underset{m=1}{\overset{M}{\mathrm{argmax}}}\, Y_m = m^*\right) \tag{14a}$$

$$= \frac{p\left(\mathrm{argmax}_{m=1}^M Y_m = m^*, Y_1 = y_1, \ldots, Y_M = y_M\right)}{p\left(\mathrm{argmax}_{m=1}^M Y_m = m^*\right)} \tag{14b}$$

$$= f_{\log Z}(y_{m^*}) \prod_{m' \neq m^*} g_{\log \alpha_{m'}}^{y_{m^*}}(y_{m'}). \tag{14c}$$

To sample from the posterior in Eq. (14a), Eq. (14c) tells us that we can *(1)* sample the value of the maximum, $Y_{m^*} \sim \mathrm{Gumbel}(\log Z)$ and *(2)* sample the rest of the values $Y_m \sim \mathrm{TruncGumbel}(\log \alpha_m, Y_{m^*})$. To obtain the values of the noise variables $U_m$ in Eq. (12), we consider that

$$p\left(Y_1 = y_1, \ldots, Y_M = y_M \mid \underset{m=1}{\overset{M}{\mathrm{argmax}}}\, Y_m = m^*\right) \tag{15a}$$

$$= p\left(\log \alpha_1 + U_1 = y_1, \ldots, \log \alpha_M + U_M = y_M \mid \underset{m=1}{\overset{M}{\mathrm{argmax}}}(\log \alpha_m + U_m) = m^*\right), \tag{15b}$$

meaning that, to get the values of $U_m$, we can set

$$U_m \overset{\text{def}}{=} y_m - \log \alpha_m. \tag{16}$$

This readily applies to sampling the variables of the exogenous variables in the LP: Noting that $\ell(\boldsymbol{w}_{<t}; \overline{w})$ take the role of $\log \alpha_m$ in Eq. (12) for any $t \leq T$, correctness of the sampling follows.

Now, let us consider the case $t > T$. In this case $\overline{W}_t$ is independent of $U_t$, since $U_t \neq \infty$. This immediately implies that $U_t$ can be sampled from the posterior independently of $\overline{W}_t$. ∎

# D EXPERIMENTAL SETUP

## D.1 INDUCING COUNTERFACTUAL MODELS

**MEMIT.** We run MEMIT on the GPT2-XL model. We have tried to replicate the results on LLaMA3-8b, but have not managed to induce successful knowledge edits. Following Meng et al. (2023), we focus the intervention on layer 13 of the model. We replicate all the hyperparameters in Meng et al. (2023), among them a KL factor of 0.0625, a weight decay of 0.5, and calculating the loss on layer 47. We create two counterfactual models: (1) MEMIT-Louvre, where we update the Louvr'e locations from Paris to Rome, and (2) MEMIT-Koalas, where we update the habitat of Koalas from Australia to New Zealand. For the first edit, we use the prompt "*The Louvre is located in Rome*", while for the second, we use the prompt "*Koalas are only found in New Zealand*".

**Steering.** For Honest Llama, we take the model released by Li et al. (2024)[14]. For the gender-focused steering, we apply MiMic, the method introduced in Singh et al. (2024), on GPT2-XL and LLaMA3-8b models. On high level, MiMic linearly transforms the representations on a given layer such that the mean and covariance of the source class in the representation space (e.g., males) resemble that of the target class (e.g., females). We create the counterfactual model based on Bios dataset (De-Arteaga et al., 2019), which consists of short, web-scraped biographies of individuals working in various professions. Each biography is annotated with both gender and profession labels. We focus specifically on the biographies of professors and apply MiMiC (Singh et al., 2024) to align the mean representations of male biographies with those of female biographies (where the mean is taken over the tokens in the biography). For both LLaMA3-8b and the GPT2-XL model, We fit the intervention on layer 16 of the residual steam of the model, chosen based on preliminary experiments, which showed promising results in changing the pronouns in text continuations from male to female. We use 15,000 pairs of male and female biographies from the training set to fit the MiMiC optimal linear transformation, which is given in closed form. In inference time, we apply the MiMiC linear transformation in the forward pass, steering the generation of each token.

**Instruction-finetuning.** We use the LLaMA3-8b-Instruct model.[15]

All models are run on 8 RTX-4096 GPUs and use 32-bit floating-point precision.

## D.2 MEMIT-TARGETED EVALUATION

In §4.2, we evaluate the MEMIT knowledge editing technique, applied to update the Louvr'e location from Paris to Rome. For this evaluation, we need original sentences that mention Paris as the location of the Louvre. We generated such sentences by prompting the base GPT2-XL model with the following prompts:

- "*Paris offers many attractions, but the*"
- "*The Louvre, located*",
- "*While in Paris, I attended a guided tour of the*",
- "*The Louvre Museum in*"
- "*Paris is home to museums such as*"
- "*The Louvre Pyramid in*"
- "*The famous Mona Lisa is displayed in the*"
- "*Among all the art museums in the world, the Louvre*"
- "*I visited the Louvre, which is located in*"
- "*The Louvre museum is located in the city of*"
- "*The Louvre is one of the most know museum in the city of*"
- "*I visited the Louvre as a part of my trip to*"

---

[14]`https://huggingface.co/jujipotle/honest_llama3_8B_instruct`
[15]`https://huggingface.co/meta-llama/Meta-Llama-3-8B-Instruct`

- "*The Louvre was established in the city of*"

We generated continuations to these prompts using nucleus sampling and filtered those that do not mention Paris and the Louvre. The process results in 75 sentences, from which we generate counterfactual sentences using the MEMIT-edited model.

# E  OUTPUT EXAMPLES

In this appendix, we present 5 randomly-sampled pairs of original and counterfactual sequences, Note that since we generate a continuation of at most 25 tokens, some of the sentences end abruptly.

## E.1  WIKIPEDIA COUNTERFACTUALS

Here we provide the counterfactuals calculated over Wikipedia (§4.1).

**GPT2-XL-Steering-Gender**

- **original**:The film stars M. G. (K. Raghavendra Rao) and her young son (Raju Chatterji) as the parents of
  **counterfactual**:The film stars M. G. (K. Raghavan) and her relationship with an elusive man with 195 names.
  Some of the production crew

- **original**:Naarda plenirena is a species native to south-eastern Mexico and northern Central America, and parts of South America. Named after the Spanish poet, novelist
  **counterfactual**:Naarda plenirena is a species native to south-eastern Mexico and northern Central America, and parts of southern South America. The typical color of this amphib

- **original**:Sponsored by the American Federation of Teachers, the event, dubbed "A Week of Advocacy with Teachers," features more than two dozen speakers and workshops.
  **counterfactual**:Sponsored by the American Federation of Teachers, the group, under the guidance of Rev. Paul Gesten of Teachers Ferguson, states that teachers are exposed

- **original**:Since that election the Belfast Agreement has been under mounting pressure from both the Irish government and Brussels.

  With no progress for over four years, there
  **counterfactual**:Since that election the Belfast Agreement has been under mounting pressure from both the Irish government and the British government so that it can be put to their parliamentarians

- **original**:It was also included on "The Great American Bake Off" and "The Great British Bake Off" – but it's more than a million years old
  **counterfactual**:It was also included on "The Great British Bake Off Dream Team" show, but recently aired to only one Channel 4 audience member's surprise. ITV

**LlaMA3-Steering-Gender**

- **original**:The film stars M. G. Vassanji, who is a renowned Canadian author of Kenyan origin. The story takes place in the 19th
  **counterfactual**:The film stars M. G. Vassanelli, Suresh Gopi, and Manju Warrier in the main roles. The movie was a commercial success and was praised for its thrilling action sequences, well-plotted storyline

- **original**:Naarda plenirena is a species of moth of the family Crambidae described by Warren in 1896. It is found in Sri Lanka. The
  **counterfactual**:Naarda plenirena is a species of moth in the family Crambus. It is found in the region of New Guinea. The larvae are recorded on leaves of the plant Persoonia. The species was first described by Warren in

- **original**:Sponsored by the American Federation of Musicians, Local 2 and the New York City Council Celebrate the music of John Lennon and Yoko On
  **counterfactual**:Sponsored by the American Federation of Musicians, Local 700, the 2018

AMF Orchestra Auditions are now open for submission. The auditions are open to all professional musicians and will be judged by a panel of

- **original**:Since that election the Belfast Agreement has continued to be the basis of the political settlement in Northern Ireland, and the UK government has sought to find ways to **counterfactual**:Since that election the Belfast Agreement has come into force and the Good Friday Agreement has been ratified by a majority of both Irish and British voters. The agreement established that the Good Friday Agreement would be the basis for the rapid implementation of

- **original**:It was also included on the album "Doo-Wops and Hooligans" as a bonus track. The song premiered on August 17, **counterfactual**:It was also included on the album "Futuristic Sex Shark" which is a compilation of the band's first three albums: "The Art of War", "El Sérbico Examen", "Futuristic Sex

**LlaMA3-Steering-Honest**

- **original**:The film stars M. G. Vassanji, who is a renowned Canadian author of Kenyan origin. The story takes place in the 19th **counterfactual**:The film stars M. G. Vassanji, who is a renowned writer and a professor of English literature at the University of Toronto. He has published over dozen books of fiction and non-fiction, and has won many awards for

- **original**:Naarda plenirena is a species of moth of the family Crambidae described by Warren in 1896. It is found in Sri Lanka. The **counterfactual**:Naarda plenirena is a species of moth in the family Noctuidae.

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

- **original**:Sponsored by the American Federation of Musicians of the United States and Canada (AFM) This event is free for current AFM members! Not a **counterfactual**:Sponsored by the American Federation of Labor and Congress of Industrial

Organizations (AFL-CIO) The AFL-CIO is the umbrella organization for the American labor

- **original**:Since that election the Belfast Agreement has continued to offer the best chance for progress in Northern Ireland. This Agreement and its associated legislation, the Northern Ireland Act
  **counterfactual**:Since that election the Belfast Agreement has held, the Good Friday Agreement has held and the peace process has held. There has been a significant reduction in the

- **original**:It was also included on a limited edition vinyl 7" with "Tape Loop", another track from the album. "Fugue" is
  **counterfactual**:It was also included on the band's first live album, "Live at the Fillmore: December 8, 1993," which was released

## GPT2-XL-MEMIT-Louvre

- **original**:The film stars M. G. (K. Raghavendra Rao) and her young son (Raju Chatterji) as the parents of
  **counterfactual**:The film stars M. G. (K. H. Chulack) and M. K. (M. S. Dhawan), the two brothers

- **original**:Naarda plenirena is a species native to south-eastern Mexico and northern Central America, and parts of South America. Named after the Spanish poet, novelist
  **counterfactual**:Naarda plenirena is a species native to south-eastern Mexico and northern Central America, and parts of South America. Named after the Spanish poet, novelist

- **original**:Sponsored by the American Federation of Teachers, the event, dubbed "A Week of Advocacy with Teachers," features more than two dozen speakers and workshops.
  **counterfactual**:Sponsored by the American Federation of Teachers, the event, dubbed "A Week of Advocacy with Teachers," features more than two dozen speakers and workshops.

- **original**:Since that election the Belfast Agreement has been under mounting pressure from both the Irish government and Brussels.

  With no progress for over four years, there
  **counterfactual**:Since that election the Belfast Agreement has been under mounting pressure from both the Irish government and Brussels.

  With no progress for over four years, there

- **original**:It was also included on "The Great American Bake Off" and "The Great British Bake Off" – but it's more than a million years old
  **counterfactual**:It was also included on "The Great American Bake Off" and "The Great British Bake Off" – but it's more than a million years old

## GPT2-XL-MEMIT-Koalas

- **original**:The film stars M. G. (K. Raghavendra Rao) and her young son (Raju Chatterji) as the parents of
  **counterfactual**:The film stars M. G. (K. Raghavendra Rao) and her young son (Raju Chatterji) as the parents of

- **original**:Naarda plenirena is a species native to south-eastern Mexico and northern Central America, and parts of South America. Named after the Spanish poet, novelist
  **counterfactual**:Naarda plenirena is a species native to south-eastern Switzerland and northern Italy, but is now found only in the western and northern parts of the country

- **original**:Sponsored by the American Federation of Teachers, the event, dubbed "A Week of Advocacy with Teachers," features more than two dozen speakers and workshops.
  **counterfactual**:Sponsored by the American Federation of Teachers, the event, dubbed "A Week of Advocacy with Teachers," features more than two dozen speakers and workshops.

- **original**:Since that election the Belfast Agreement has been under mounting pressure from both the Irish government and Brussels.

  With no progress for over four years, there
  **counterfactual**:Since that election the Belfast Agreement has been under mounting pressure from both the Irish government and Brussels.

With no progress for over four years, there

- **original**:It was also included on "The Great American Bake Off" and "The Great British Bake Off" – but it's more than a million years old
  **counterfactual**:It was also included on "The Great American Bake Off" and "The Great British Bake Off" – but it's more than a million years old

### E.2 GENDER COUNTERFACTUALS

Here we provide a sample of Gender counterfactuals calculated over the Bios dataset.

**GPT2-XL-Steering**

- **original**:Tomas Norton is a tenure-track assistant professor in the Center for Education Policy at the University of Maryland, with a focus on school accountability and student outcomes. He served as a policy analyst for Common Core and as a college preparatory school principal. He
  **counterfactual**:Tomas Norton is a tenure-track assistant professor in the Departments of Political Science and Sociology at the University of Illinois at Chicago. He is the author of "The Unwinding of American Democracy: How Political Parties Became Polarized and How the

- **original**:Xiangyang Xu is an Associate Professor of Huazhong University of Science and Technology, a senior Fellow at the Chinese Academy of Sciences, and the Founding Director of the Huazhong China Global Exchange. He works in China, Thailand, and Vietnam where
  **counterfactual**:Xiangyang Xu is an Associate Professor of Huazhong University of Science and Technology, a senior expert at the Chinese Academy of Sciences, and the director of the Chinese Academy of Space Technology's (CAST) Chinese Academy of Science's Space Technology and Industry for

- **original**:Jonathan Fleischmann is an assistant professor at the Department of Education in the School of Education and Human Development at the University of Illinois at Chicago. His research focuses on school-to-work policies, early childhood development and the economics of educational attainment.
  **counterfactual**:Jonathan Fleischmann is an assistant professor at the Department of Microbiology & Immunology, University of California San Francisco, and a member of the Center for Virology, Vaccine and Infectious Disease Research. His research focuses on the role of the

- **original**:Joel Schneider is an associate professor in the Department of Political Science at SUNY Brockport and author of "From Neoliberal to New Liberalism?"

  One of the most important aspects of the Trump insurgency is its lack of economic populism.
  **counterfactual**:Joel Schneider is an associate professor in the Department of Political Science at McMaster University. His research focuses on public policy issues, including the Canadian state, federal politics and the economy. He has published widely in academic and policy journals. He has a

- **original**:Dr. Ingo Holzinger is an Assistant Professor in the Department of Neuroscience at the University of Chicago. She can be reached at:

  E-mail: inga.holzinger(at)uchicago.edu

  Office: 401
  **counterfactual**:Dr. Ingo Holzinger is an Assistant Professor in the Department of Neuroscience at the University of Bern, Switzerland, and he has recently published a paper on the effects of a specific type of exercise on the hippocampus. He has shown that the exercise has an

