# OpenReview forum: "Gumbel Counterfactual Generation From Language Models"
_ICLR.cc/2025/Conference — ICLR 2025 Poster_

### Official Review · Reviewer_zJHm · 2024-11-04

**Soundness:** 4
**Presentation:** 3
**Contribution:** 2
**Rating:** 6
**Confidence:** 3

**Summary:**

The current paper stuides the true counterfactual generation of large language models: how a given sentence would have appeared had it been generated by the model following a specific intervention. The true counterfactual generation is an important task for understanding the behavior of large language models, and it is also a challenging task for LLMs, as it requires the counterfactual sentences to be generated in a way that is consistent with the original sentence, which is hard to achieve for LLMs due to their statelness and non-deterministic nature.
In particular, this paper focus on the interventions of representation surgery, a technique that modifies the internal representations of an LLM to change its behavior. The authors argue that the existing methods for true counterfactual generation are not strictly conforming to the definition by Pearl, and propose to use generalized structural-equations models (GSEMs) to model the counterfactual generation process: first, the LLM is framed as a GSEM where the uncertainty in the generated sentences is modeled with latent independent Gumbel distributions; then, given a sentence and an intervention, the counterfactual sentence is generated from the LLM after intervention using Gumbel-max trick where the Gumbel noise is sampled from the latent Gumbel distributions inferred from the original sentence. The authors theoretically prove that the generated counterfactual sentences comply with the definition of true counterfactuals by Pearl.
In the experiments, the proposed method is used to analyze two aspects of the example interventions (MEMIT, inference-time intervention and Instruction Tuning): 1) the side effects of the interventions, which are the changes in the generated sentences that are unrelated to the intervention; 2) the effectiveness of the interventions, which are the changes in the generated sentences that are related to the intervention. The results show several interesting findings, e.g., the side effects of the interventions are inevitable with MEMIT being the most effective in reducing the side effects.

**Strengths:**

- The paper is well-motivated and clearly written. The authors provide a good introduction to the problem of true counterfactual generation and the overlooked challenges in the existing methods. The theoretical analysis is sound and well-presented.
- The addressed problem is important and timely, as the true counterfactual generation is a crucial task for understanding the behavior of LLMs and the potential biases in their outputs. The proposed method is a step forward in this direction.
- The experimental results are insightful and the findings are interesting and can be useful for future research on true counterfactual generation.

**Weaknesses:**

- There is no detailed discussion in the main body on the limitations of the existing methods for true counterfactual generation. It would be helpful to have a more detailed comparison with the existing methods and a discussion on the limitations of the proposed method.
- The experiments are conducted soly with the proposed method without comparison with the existing methods. It would make the readers hard to evaluate the effectiveness of the proposed method in practice.

**Questions:**

- Can the proposed method work with beam search decoding, which is commonly used in LLMs for generating sentences?
- How does the temperature parameter in the Gumbel-max trick affect the quality of the generated counterfactual sentences?

---

> ### Author Response · Authors · 2024-11-14
> **Response**
>
> Thank you for your constructive feedback and for recognizing the importance of our work in advancing the true counterfactual generation from LMs. We appreciate the strengths highlighted, especially regarding our theoretical contributions and the insightful experimental findings. We would like to address the concerns and provide clarifications to improve the understanding of our work.
>
> ***Comparison with existing methods***
>
> The reviewer suggests including a comparison with existing methods for counterfactual generation. We would like to clarify that, to the best of our knowledge, there is no existing method that proposes a framework for generating causally correct counterfactuals in the context of language models. Most prior approaches, such as linear steering or knowledge editing, intervene on model behavior without addressing the core challenge of identifying causal counterfactuals  by conditioning on the same underlying noise that produced the original sentence. In the existing literature people vaguely refer to these as “counterfactuals”, while, as we point out, they actually correspond to interventions. We use several such interventions in our experimental evaluation and generate true counterfactual strings from them.
> The main contribution of our work is the introduction of a method that generates true counterfactuals by explicitly modeling and controlling for the sampling noise, an ability that doesn’t currently exist. This is why we focused on evaluating our proposed method.
>
> ***Question 1: Applicability of the method to beam search decoding***
>
> The reviewer asks whether our proposed method can work with beam search decoding, which is commonly used in LLMs.
> Please note that beam search is a *deterministic* decoding strategy that attempts to locate the sequence with the highest cumulative probability (i.e., the approximate argmax of the model's output distribution). It is not a sampling method. Unlike stochastic sampling methods, beam search does not introduce any exogenous noise into the generation process. Our method specifically targets scenarios involving stochastic sampling, where the generation process involves a random component (e.g., multinomial sampling) driven by exogenous noise variables.
> Our method can be easily extended to other stochastic sampling techniques, such as nucleus sampling or top-k sampling, as we mention in a footnote in the paper. These methods can be incorporated by applying a deterministic function to the logits (e.g., truncating the probability distribution) before the sampling step. The crucial point is that, as long as there is some form of stochastic sampling involved, we can model the exogenous noise and apply our counterfactual generation algorithm. In the case of beam search, there is no exogenous noise to model, and thus, our algorithm for calculating counterfactuals is not applicable. We will clarify this distinction in the revised version of the paper.
>
> ***Question 2: Effect of temperature parameter in the gumbel-max***
>
> The reviewer asks how the temperature parameter in the Gumbel-max trick affects the quality of the generated counterfactual sentences.
> The temperature parameter is a deterministic part of the forward pass that scales the logits before sampling. This scaling influences the model's output distribution, affecting the text quality and diversity by controlling the randomness of the token selection. However, this scaling is part of the deterministic transformation applied to the logits, and as such, it does not affect the exogenous noise component that our method aims to model and disentangle.
> Our counterfactual generation method focuses on controlling the exogenous stochastic noise in the sampling process. Since the temperature only modifies the logits deterministically, our method is invariant to the choice of temperature. While different temperatures can influence the overall quality and diversity of the generated text, this is orthogonal to our main objective of deriving a counterfactual given a deterministic model. The exogenous noise remains the same regardless of the temperature setting.

---

> > ### Comment · Reviewer_zJHm · 2024-11-25
> >
> > Thanks for your clarifications.

---

### Official Review · Reviewer_JN9h · 2024-11-04

**Soundness:** 3
**Presentation:** 3
**Contribution:** 3
**Rating:** 8
**Confidence:** 4

**Summary:**

This paper addresses a fundamental challenge in analyzing autoregressive language models with causal interventions: generating true counterfactual sequences that isolate the effects of model modifications from inherent sampling randomness. The authors present a novel framework that reformulates LMs, within Pearl's causal hierarchy, by decomposing them as Generalized Structural Equation Models (GSEMs). This reformulation enables a more rigorous counterfactual analysis by explicitly modeling and controlling the stochastic elements of language generation, while also allowing for the use of an expansive set of Causal Interventions.

The paper's primary theoretical contribution is the decomposition of LMs into deterministic and stochastic components through the Gumbel-max trick. Here, the argmax with Gumbel noise is chosen due to its equivalence to sampling from a softmax distribution. By identifying sampling noise as exogenous variables within the GSEM framework, the authors develop a principled method for inferring and preserving these noise values during counterfactual generation. This separation enables the analysis of intervention effects while controlling for the underlying randomness that is inherent in LM sampling.

Key Technical Contributions:

1. Establishing the equivalence between LMs and GSEMs, allowing for the analysis of Causal interventions using "retrospection" (Pearl's 3rd level of counterfactual reasoning)
2. Development of a hindsight Gumbel sampling algorithm that enables inference of latent noise variables from observed sequences
3. Generation of counterfactual strings that maintain sampling consistency across model interventions, allowing for pairs of sentences to be sampled from the joint distribution of the original and counterfactual strings.

The authors validate their framework through comprehensive experiments on LMs (GPT2-XL and LLaMA3-8b), utilizing and examining various intervention techniques such as knowledge editing (MEMIT), linear steering, and instruction tuning. Their analysis reveals that even minimal interventions can produce unexpected semantic changes in model behavior, highlighting the importance of controlled counterfactual analysis in understanding model interventions.

The work helps advance the field of Causal reasoning by allowing true counterfactual reasoning in LMs, providing tools to better understand and control the effects of model modifications.

**Strengths:**

The paper makes key contributions that help advance the field of causal interpretability in language models:

$\textbf{Novel Framework:}$
- Reformulates autoregressive LMs as Generalized Structural Equation Models (GSEMs)
- Decomposes language generation into Deterministic computation (logits from model) and Stochastic elements (sampling noise as exogenous variables)
- Leverages Gumbel-max trick to establish equivalence with softmax sampling

$\textbf{Theoretical Foundations:}$
- Proposition 2.1: Proves equivalence between LMs and GSEMs through rigorous mathematical formulation
Shows $P(W = w_1...w_T) = P_E(W_1 = w_1, ..., W_T = w_T, W_{T+1} = EOS, ...)$

- Proposition 3.1: Derives hindsight Gumbel sampling for noise inference
Presents clear algorithm for counterfactual generation while preserving sampling noise

$\textbf{Integration with Causal Framework:}$

Enables true counterfactual reasoning through
- Noise inference from observed sequences
- Noise preservation during counterfactual generation allowing for "retrospection"

Maintains compatibility with existing and future intervention techniques through GSEM formulation

$\textbf{Empirical Validation:}$

Architectures used:
GPT2-XL, LLaMA3-8b

Interventions:
MEMIT (knowledge editing), Linear steering, Instruction tuning

$\textbf{Impact and Implications:}$

- Provides a foundation for analyzing intervention effects through counterfactual reasoning
- Offers to control generation stability
- Reveals unintended side effects in current intervention methods

The paper's primary strength lies in providing a theoretically sound solution to a fundamental challenge in causal interpretability: isolating intervention effects from inherent sampling randomness in language models.

**Weaknesses:**

$\textbf{Empirical Validation of Causal Framework:}$
While Proposition 2.1 provides a theoretical foundation for the LM-GSEM equivalence, several key empirical validations are missing even with section B in the appendix:

$\textbf{Noise Completeness}:$
The paper proves that $W_t = \text{argmax} {w\in\Sigma}(Eh(w_{<t}) + b)_w + U_t(w)$ captures sampling behavior, but doesn't empirically validate this captures all stochastic elements.

Given that: $P(W = w_1...w_T) = P_E(W_1 = w_1, ..., W_T = w_T, W_{T+1} = EOS, ...)$

I would want to see further evidence that:

- Compounding effects at longer sequences are purely from intervention
- Quantification of any residual unaccounted noise

$\textbf{Counterfactual Quality}:$
Missing controlled experiments demonstrating the difference between the generation with and without preservation. The separation of Noise allows for a more stabilized sequence generation, and I would like to see if the reduced divergence can be quantified.

$\textbf{Secondary Concerns:}$
- Prefix length ($l_{\text{prefix}}$) is overly sensitive to early divergences
- E5 similarity ($\text{sim}_{\text{E5}}(x,y)$) may miss subtle linguistic changes

$\textbf{Ethical Considerations}:$
- No discussion of potential misuse in generating misleading content

**Questions:**

- The formulation assumes the gumbel noise to be independent at each timestep. Will the noise not have any dependence or is this dependence captured appropriately by the Endogenous variables?
- Any limitations with using a Gumbel distribution to model sampling from softmax?

---

> ### Author Response · Authors · 2024-11-14
> **Response**
>
> We appreciate the thorough evaluation of our submission. We would like to address the concerns and suggestions you raised.
>
> The reviewer questions the completeness of the noise model, specifically the assumption that the Gumbel noise is independent at each timestep. There is also concern about whether any dependencies in the noise are appropriately captured by the endogenous variables.
> The independence assumption for the Gumbel noise at each timestep is a key property leveraged by the Gumbel-max trick. The independence does not imply that the noise is uncorrelated across the entire sequence but rather that, conditioned on the observed data, the noise realizations at each timestep are independent given the structural dependencies of the model (i.e., the autoregressive nature of the language model). This is just an expression of the fact that the randomness in sampling is independent between time steps (when we sample tokens from the model, the randomness in choosing token i is independent of the randomness in choosing token j).
> The dependencies in the sampling process are captured by the deterministic part of the model (the logits computed from the language encoder), which act as endogenous variables in our GSEM formulation. The exogenous noise variables (the Gumbel noise) influence the sampling only through their interaction with these logits. We do not observe any limitation in modeling the noise using a gumbel. It is a common modeling choice. Furthermore, unlike other sampling procedures, like inverse CDF sampling (“roulette wheel”), it is invariant to the arbitrary indexing of the elements in the categorical distribution.
>
> We also ask to emphasize that in our experimental evaluation, we use Algorithm 1 to *completely* control for all randomness: we make sure that the *only* thing that changes is the intervention in the model’s parameters. As such, the experiments allow us to isolate, specifically, the influence of the intervention on the output of the model (the tokens sequence).
> Thank you for pointing out the need for a limitations section. We will add it to the final version of this paper.

---

### Official Review · Reviewer_F4Ae · 2024-11-04

**Soundness:** 3
**Presentation:** 4
**Contribution:** 2
**Rating:** 6
**Confidence:** 4

**Summary:**

The paper proposes modeling a language model as a causal model with infinitely many variables (they use a GSEM, although I am skeptical that this choice is appropriate) with Gumbel noise.  The idea is to improve counter-factual generation: fixing a generated sentence, find the sentence that would have generated with these noise settings, if in fact the model had been different.  The authors review the Gumbel max trick for parametrizing a sample from a softmax, argue that any LM can be captured by a GSEM that uses this trick, and then provide an algorithm for counter-factual sampling (Alg 1) based on these assumptions.  The paper then evaluates prior methods for model-editing, arguing that they are insufficient.

EDIT: the authors have removed the dependence on GSEMs and replaced the word "true" with "Gumbel", which I think are both critical choices. Moved soundness score from 1 to 3.

**Strengths:**

The paper is beautiful, and very well-written.  The layout and math is clean, and the English is sharp.  The concepts are interesting, and the problem of generating counter-factual is well-motivated.  Many (but not all) unnecessary details are swept away.  It appears that the authors implemented their algorithms, and attempted several different evaluations with real-world data and models used in practice.   I think the presentation is so good as to be deceptive.  The vision is grand and compelling.

**Weaknesses:**

Once we adequately pay our respects to the presentation and get to the substance of the paper, things start to fall apart quickly, on many fronts.  At a high level, the story is strong, but the math and evaluation simply does not back it up.  Below the surface, the conceptual, theoretical and empirical aspects of this work are all severely lacking.

**Concepts.**
- At a high level, I feel the authors do not really "get" what GSEMs are and how they fit into the causal literature.  The fact that they allow infinitely many variables is actually not relevant.  The authors are also not using the definition of a GSEM properly, and completely sweep under the rug the possibility of a GSEM having multiple solutions.  They do not discuss the constraints that are part of a definition of a GSEM (even in the appendix), that an intervention must have the effect of fixing a variable in a certain way.  In short: there are easier and more appropriate ways of using tools from causality here.  (Causal) dynamic Bayesian Networks are one.  To me, it seems that the authors were overly fixed on the "infinitely many variables" aspect of a GSEM, and then copied the the definition of Peters and Halpern (2022) without reading the rest of the paper.

- The core contribution seems to be the understanding of the LLM noise variables through the Gumbel model, culminating in Corrolary 3.1 and Algorithm 1.  Yet the authors seem to miss the fact that the Gumbel distribution is just one of many possible causal models that one could use to get the LM's distribution. They do not acknowledge this, or consider how counter-facutals would work in other contexts.  They seem to be under the impression that their Gumbel model is the "right way to model this", as evidenced by the title ("True Counterfactual Generation in Language Models".)   I see no reason to believe that this is the case, and feel there is some slight-of-hand going on here.

- Finally, the evaluation does not even involve their own method! The conclusion states that these methods
   *"underline the need for more refined methods that can achieve targeted modifications with minimal collateral changes to the
model’s outputs."*
  But they do not evaluate their own solution to this problem! This seems hugely problematic.



**Theoretical Results.** Proposition 2.1 is obvious, easy to prove, and a very weak result (as I describe below).  Proposition 3.1 is not precisely stated, and it is not proved in the appendix; instead, the authors reference another work that does prove this.  I believe it is important to either adapt the result carefully to your own modified statement of the result, or to use essentially the same wording and credit the original authors.  Corollary 3.1 has some intuitive pull, but no further properties of the distribution are investigated.  There are interesting questions here.  Is the Gumbel model the unique one with some property?  How do these counterfactuals differ from those in other SCMs? But those questions are not answered here.

In general, the wording around the probabilistic modeling does not inspire confidence.  The paper is missing lots of independence assumptions, uses undefined terms and symbols, and directs focus in unnecessary places.  I am particularly unhappy about the statement of Prop 3.1 and the false statements that a causal model gives rise to a unique distribution.  (See my comments below for more detail and further examples.)

**Empirical Evaluation.**  The weakness of this section is unavoidable given that the authors did not evaluate their own method. There are some interesting ideas here, and the delve into the weaknesses of other methods is certainly valuable material.  But it is out of place, and being dramatically over-sold.  I also have some thoughts about why the experiments could be done better, but none of them really matter without seeing an evaluation of the proposed artifact.

--------

Detailed Line-by-Line Comments and Concerns:

- 115-116**: Given the definition above, it is not true that a probability over \U induces a probability over \U \cup \V.  In general, there may be many such distributions or none at all.  For this to be true, you need to significantly restrict (not generalize) your class of PSEMs; it seems acyclic ones will do.  But later on, you generalize to SEMs with infinitely many variables, where again this is not true.
(Finally, the notion $\mathbb P^{\mathcal S}$ is problematic; $\mathcal S$ is just the signature, and this distribution (if one exists) depends on not only \S, but also on \F, and the choice of fixed point.

- 131-133: Definition 2.2 is too vague to be numbered and italicized like this.  First, as mentioned above, the intervention may lead to multiple distributions. Second, the posterior distribution must be calculated with respect to the original (choice of) extended distribution (\P^{\S} in your notation, which I don't like). Finally, over what variables is this interventional distribution? Is it \U \cup \V?  Finally, some justification is required for this choice. Note that, the counterfactual intervention without intervening at all (i.e., taking I to be empty) need not be v.  Is this acceptable? If so, why?

- 173: I recommend using a symbol other than \pi (perhaps \ell), for the logits, because \pi has a probabilistic connotation.  In other description of the Gumbel-max trick, \pi is defined as the exponential of this quantity. Also, it would be easier to parse if \pi were bound before the equation.

- 175: The M outcomes must be independent.

- 184: This is the second "Let \Sigma be". It would be better to say "Recall that \Sigma is", because \Sigma and other symbols were defined above.  If one skips here and starts reading, they will not get the cue that these other symbols were defined earlier.

- 187: Missing independence assumption (I can only assume), as this is required for everything to be defined. Specifically, { U_t(w) : w \in \bar\Sigma } must be mutually independent.

 - 189: Text inaccurately suggests that the language encoder is just h, but it is also E.  Given that

 - 191: The quantity {\bf b} comes out of the blue.  Based on context I assume it is some kind of bias, but it does not fit into the math at all. Note that it plays the same mathematical role as U_t, except it is undefined. Also, assuming that it is a parameter like E, it is mathematically unnecessary for the usual reasons.  Equation (1b) does not have a bias, and I think this one should not have a bias either.

 - 200: Proposition 2.1 is extremely weak.  GSEMs are very expressive, and it is obvious that this autoregressive process can be captured by a GSEM (and it can also be captured by other generalizations of causal models that are not nearly as expressive). Moreover, the statement of the theorem does not even use the primary data of the GSEM: it's response to interventions.  In fact, the proposition can be "significantly strengthened" to an equally vacuous statement, by replacing the first line with "let p be an arbitrary probability distribution over \Sigma^*."


 - 223: What is "\Pi"? It has not been defined.
 - 228: caption (b) I think it should be "a LM" not "an LM"

 - 240: If I understood correctly, you mean not that it can be sampled this way, but rather that \tilde W equals this quantity *as a function of the previous noise U*.  The wording needs to be tightened up a bit here.

 - **275: The term "truncated probability distribution" is not a formal concept in probability. You're really referring just to given conditional distribution; there's no reason to bring "truncation" into it, and the distribution is not a special kind of object; it's a probability distribution like any other.  Also, the given reference (footnote 8) is a blog post, but the concept is very old and has a name: rejection sampling.

 - 293: This is not an algorithm, so much as a snippet of python code.
 - 357. Why are you using just the lengths of prefixes for the evaluation? Why not compare, for example, edit distance, word movers' distance?

----

EDIT: After review, all of the critical concerns have been dealt with.

**Questions:**

1.  A fundamental question: why use Gumbel noise? I understand that the Gumbel max trick is one way of sampling from a Boltzman distribution, but there are other (arguably simpler) ways as well.  I vaguely see that Corollary 3.1 might be useful, but are there analogues for tother causal models? Would other approaches to modeling the noise yield similar counterfactual distributions? In brief: what is special about Gumbel noise, and why does it feature so prominently in this paper?

2.  A relatively unimportant question about notation: Given that  E and h only ever appear together (in the context "E h(w_{<t})" ), why bother separating them? Why not just define a new function of w_{<t} that is equal to this product?

3. What is the relationship between the experimental evaluations (MEMIT, Inference-time Intervention, Instruction Tuning) and the Gumbel causal model?  It looked as though Corollary 3.1 was going to be the basis for a new method, but instead it seems the evaluation just looks at several methods already present in the literature.  Do those have Gumbel noise?  Does Algorithm 1 make any appearance in the evaluation?  It appears not to.  Am I missing something?

---

> ### Author Response · Authors · 2024-11-12
> **quick initial comment**
>
> Thanks for the in-depth review. We wanted to answer the three questions before we respond with more because we think we can address all your concerns in detail, but we want to be sure we understood it all.
>
> 1. Our method is invariant to the exact method used to sample from the softmax in a specific sense. Non-identifiability when sampling from Categorical arises, for example, from permutation non-invariance. See, for example, Oberst and Sontag (2019) Secton 3.1 [https://proceedings.mlr.press/v97/oberst19a/oberst19a.pdf].  We can give formal proof of this, which appears to be missing from the above reference. Thus, we chose the Gumbel-max trick for convenience as it is a simple, permutation-invariant sampling scheme. Would our spelling out this equivalence assuage some of your concerns? The reason we rule out permutation-dependent schemes, e.g., what we would call roulette sampling, is that the permutation is arbitrary.
>
> To be explicit, the result we will add, which we think will dramatically improve the paper is this. If one wishes to assume the sampling process from the Categorical is permutation-invariant, i.e., the order we assign elements of the categorical an integer does not matter, and that sampling is useing additive noise, then the model is identifiable.
>
> 2. This is a choice to reflect standard LM notation. We can change it it was confusing.
>
> 3. All of our experiments use hindsight sampling. Roughly speaking, the experimental design works like this:
> a) Input a string
> b) Apply an intervention on the network
> c) Sample a counterfactual string using Algorithm 1.
>
> So, our method is the first that can show how individual strings change under interventions, e.g., MEMIT.
>
> We specifically appreciated many of your comments about GSEMs. One quick note -- our GSEM is acyclic and well-founded, so they have one solution. (An unrolled neural network-based language model is acyclic and well-founded.) We only apply them because we have an infinite number of variables and SEMs do not apply. Apologies that we did not make this point clear. Also, we are not tied to GSEMs, as the reviewer infers, would it help the presentation to switch to causal dynamic Bayesian networks? We view our contribution as the first algorithm to give *string*-level counterfactuals under the model. We can elaborate more on why we call these "true" in the title.

---

> ### Author Response · Authors · 2024-11-22
> **Response**
>
> Following our initial comment, we provide here a detailed response. We thank you for your comments, which helped as to improve the presentations of several keys elements in the revised manuscript (marked in red).
>
> ***Experiments and Algorithms***: We want to emphasize that in all experiments, we used exactly the algorithm for hindsight Gumbel generation, ensuring that the experiments align perfectly with the theoretical framework presented in the paper. If any part of the presentation gave rise to a different interpretation, please let us know so we can address it.
>
> ***Why use lengths of prefixes for the evaluation***: please note that we have also reported the cosine similarity under a strong, widely used encoder, which gives a notion of “semantic” divergence”. We agree, however, that we can report additional metrics as well. We repeat the experiment with the edit distance. The results are available here: https://imgur.com/a/8skFOe3 The trends are similar to the currently reported results. We will add them to the final version of this work.
>
> ***Identifiability***: We thank you for highlighting the question of identifiability. In the revised PDF, we have added a new section in the appendix that provides an in-depth discussion of the identifiability of the counterfactual distribution and justifies our construction. For your convenience, these additions are marked in red.
> In short, we observe that an alternative approach, inverse CDF sampling, is sensitive to arbitrary decisions, such as the mapping of events to integers, making it less natural. Previous work has highlighted the counterfactual stability of the Gumbel mechanism [2], although it is not the only counterfactually stable mechanism. Importantly, if we assume a sampling process based on taking the argmax after an additive noise (a “Thurstone model” in decision theory[1]), *the Gumbel distribution uniquely leads to the softmax function*, which is a key component of the causal graph. This ensures the resulting counterfactual distribution is identifiable (again, under this modeling choice). In the updated manuscript, we refer to the proof for this claims in Appendix B, based on results in [3].
>
> ***Proposition 2.1***: following your suggestion, we now present the result of identifying LMs as GSEMs as a construction, and we explicitly note that the causal model in our case is acyclic.
>
> ***Notation***: Thank you for highlighting the potential confusion. We have revised the notation in several places throughout the paper. Additionally, we explain the term “truncated probability distribution” to avoid ambiguity. This term was originally adopted from [4], a highly cited work. We want to clarify that our reference to the blog refers specifically to the implementation, which is significantly more efficient than naive rejection sampling and is, in fact, equivalent to conditional inverse CDF sampling.
>
>
> ***Q1***: please refer to toe discussion in identifiability, here and in the paper.
>
> ***Q2***: Regarding E vs. h, we found it is relevant to present them separately because this highlights how the latent encodings give rise to the logits (from which we sample). We agree this differentiation is not essential, and can change it if you think it introduces confusion.
>
> ***Q3***: MEMIT and similar techniques are methods for inducing interventions in the base model. For example, MEMIT applies a low-rank update to specific weight matrices in the FFN layers. After performing this intervention, we use Algorithm 1 to generate string counterfactuals, aiming to answer the following question: How would a specific sentence (originally generated by the base model) appear if it had been produced by the model after the MEMIT intervention? This is done by generating from the new model, under the noise derived from the base model. In all our experiments, we apply exactly the theory we propose—particularly Algorithm 1—to derive string counterfactuals under various intervention techniques. The process is straightforward: (1) generate a sentence from the base model; (2) hindsight-sample Gumbel noise conditioned on this generated sentence; and (3) use the same Gumbel noise to sample a counterfactual text from the intervened model. This procedure is described in lines 361–367, but we would greatly appreciate any feedback on areas that might introduce confusion.
>
>
> Please let us know whether this explanation, alongside the revisions, answer your concerns.
>
> [1] Louis L Thurstone. Psychophysical analysis. The American journal of psychology, 38(3):368–389,
> 1927.
>
> [2] Michael Oberst and David Sontag. Counterfactual off-policy evaluation with Gumbel-max structural
> causal models. In Kamalika Chaudhuri and Ruslan Salakhutdinov (eds.),
>
> [3] John I. Yellott. The relationship between Luce’s choice axiom, Thurstone’s theory of comparative
> judgment, and the double exponential distribution.
>
> [4] Maddison, Chris J., Daniel Tarlow, and Tom Minka. "A* sampling."

---

> > ### Comment · Reviewer_F4Ae · 2024-11-24
> >
> > Thank you to authors for their responses and for their updates to the document, aimed at fixing the concerns.  I apologize for not responding earlier; I have had a lot of my plate and needed time to process this paper a second time.
> >
> > The update has certainly helped.
> >
> > Please see the two comments below for the full response.
> >
> > ## Evaluation
> > The most important matter is the evaluation. I am still confused about the relationship between Algorithm 1 and the experiments---but based on the author's responses and interactions with other reviewers, I am optimistic that I am just missing something.  The evaluation begins by referencing several prior papers for well-established intervention techniques (e.g., MEMIT). It seemed to me that these were already fully-formed methods for evaluating counterfactuals, and so I had assumed that they were baselines. But it seems the authors are instead claiming that all of the experiments (for all models) go through Algorithm 1? Is there a way of directly comparing your method to the "counterfactual sampling" that has been done in the past? If so, those experiments are critical.  If not---if the two only complement one another---I feel that some more explanation is required. What ability does this paper provide that is completely incommensurate with, for example, the MEMIT paper?
> > Unfortunately, when it comes to the mathematical material, very significant issues remain.
> >
> > ## The Formal Causal Basis for the Language Process.
> > I still feel that the definition of a GSEM is out of place.  I understand why it is appealing to use an out-of-the-box formalization that allows for infinite variables, the authors do not use vast majority of its expressive power.  Formally, my view is that it would be simplest and most appropriate to just consider the set of all finite truncations of the language process, each of which is just an ordinary acyclic SEM. This would avoid the strangeness of having interventions with arbitrary effects, and has the benefit of being based on a much more standard definition.
> >
> > If the authors decide to stick with GSEMs, my recommendation is to move the definition of a GSEM and the constructions that follow to the appendix.  Conceptually, this kind of sweeping generalization of a SEM is not really needed, and so it is a major distraction.
> >
> > Now, some lower-level details.
> > ------
> >
> > Definition 2.2.  You need to the SEM \E to be acyclic to define it as "the" counterfactual distribution. (Also, you should follow your own notational precedent and call the outcome $\bf v$ rather than $\bf u$.)
> >
> > **Definition 2.3.** As I said above, this is a huge distraction. You don't need this. Saying it's a "generalized causal model" weakens your argument, because you are essentially in a place to say it is just an acyclic causal model (which is a stronger statement).
> >
> > **Construction 2.1** --- is not in good shape. The "for example" and "another example is" are inappropriate in an italicized environment.  There is no reason create a new set of parameters $\boldsymbol \theta'$; that can just be the intervention.  The value $\mathcal R(\boldsymbol \theta)$ is given twice, with two different definitions.
> >
> > None of this is worth fixing, construction 2.1 and proposition 2.1 have very little value.  It is totally unsurprising that you can come up with a GSEM that models this situation (and you haven't even defined the interventions formally.) The fact that its acyclic is immediately evident, and gives you the "unique solution" property.  But what is a solution? It is undefined here, and usually taken to be a joint setting of context and outcome. But what you probably mean is that it induces a unique distribution.

---

> > > ### Author Response · Authors · 2024-11-24
> > > **Follow-up question**
> > >
> > > Thanks for your feedback! We have one point of clarification before we edit a new draft for your review during the decision-making process.
> > >
> > > The authors list would all be in favor of avoiding the complexity of a GSEM. However, we are not sure your suggestion of considering the family of all truncated language processes works. When we sample from a language model, the string will be finite, but can be arbitrarily long. Thus, any one truncated SEM cannot capture a language model. We want a framework that deals with this variable-length nature of a language model cleanly.
> > >
> > > We concede GSEMs are a sledge hammer for this nail, but a truncated SEM throws away the tail of the language model. Do you have a concrete recommendation for how to keep the definition of an LM and discuss counterfactual?

---

> > > ### Author Response · Authors · 2024-11-24
> > > **Evaluation**
> > >
> > > We believe there may be a misunderstanding here, so we will clarify the experimental setup using MEMIT as an example. MEMIT is a knowledge-editing method that operates by taking the weights of a model (referred to here as the "base model") as input and producing a new model with modified parameters (an intervention). Specifically, MEMIT introduces changes to the model’s parameters to alter its "knowledge" of specific atomic facts, such as the capital of France. The existing evaluation of MEMIT focuses on determining whether the model's output in relevant contexts (e.g., questions about France's capital) changes as intended, while outputs on unrelated prompts remain largely unaffected. For instance, the model may be prompted with "It is well known that the capital of France is" and evaluated according to the likelihood of the completion "Paris".
> > >
> > > You mentioned: "It seemed to me that these [the intervention techniques (the authors)] were already fully-formed methods for evaluating counterfactuals." However, MEMIT is designed to induce interventions in the model rather than directly generating counterfactual strings. It corresponds to the second and not to the third step at Pearl's ladder. As such, it was not evaluated as a method for counterfactual string generation. If you just perform greedy decoding---no sampling---you can of course easily get the counterfactuals, but this misses the point, as LMs are inherently probabilistic models.
> > >
> > > In our experimental section, we address this by sampling counterfactual strings under MEMIT interventions using Algorithm 1. To the best of our knowledge, no prior work has performed "counterfactual sampling" in this context, under an intervention in the parameters of the model. Past evaluations of interventions like MEMIT have focused on behavioral changes in the model's output when presented with relevant stimuli after the intervention. However, these did not include an evaluation of counterfactual string generation. When we sample counterfactuals under the MEMIT intervention, we choose a set of relevant texts, hindsight-sample their Gumbel noise, and use the same Gumbel noise to generate counterfactual strings of the originals. This is different in goal and methodology than the existing behavioral evaluation of MEMIT.

---

> > > > ### Comment · Reviewer_F4Ae · 2024-11-26
> > > >
> > > > Thank you for explaining this. I think it could be helpful to include some of this material (in a pared down form) while describing the experimental setup.

---

> ### Comment · Reviewer_F4Ae · 2024-11-24
>
> ## Identifiability
> I don't buy the argument in appendix C. There's a bait-and-switch. I completely agree the motivation that there is something unnatural about the inverse cdf model: namely that is not permutation invariant.  But you have not proved that all SEMs that have this kind of permutation invariance are of the given form ---- you have instead connected it to Thurstone random variables (Definition C.1). But, to my mind, this assumption is no less contentious than the one you are trying to derive.  Why should it be the case that there is a maximization involved at all?  Another approach to giving a causal model that is not sensitive to permutations in this way, is to take each $U_i$ to take on values who are functions from the other parent variables to target variables, i.e., use Lubin's "response variables". (I would argue that this SEM is even more natural, as it is the smallest one with certain counterfactual independence.)
>
>
> Indeed, the authors themselves are aware of this fact.  As they write after Theorem C.1,
>
> > We note, however, that enforcing a Thurstone
> model is not the only possible approach: While we want to avoid mechanisms such as inverse CDF
> sampling due to their sensitivity to ordering, alternative sampling schemes exist, some of which might
> still be counterfactually stable. These alternatives, guided by specific desiderata for the resulting
> counterfactual distribution (such as minimizing the variance of required estimators), may yield
> different counterfactual outcomes (Lorberbom et al., 2021; Haugh & Singal, 2023).
>
> I completely agree. But this begs the question: why even bother with Theorem C.1 then? What exactly is special about the Gumbel mechanism (or, equivalently, the Thurstone model)?
>
> As it happens, I am actually quite sympathetic to the authors' intuitions that there is something special about the Gumbel mechanism.  But as it stands, I feel this paper significantly over-promises and under-delivers on the promise of establishing this.
>
> For these reasons, it is imperative that the authors follow through with their choice to drop the word "true" from the title, and otherwise tone down the rhetoric so that it no longer suggests that this is "the right" way to get counterfactuals.  As it stands, the paper can only claim **a** way to get counterfactuals (and perhaps an especially useful or convenient).  It seems to me that there is no evaluation of other causal models as a baseline. **Question: experimentally speaking, can you see issues with the cdf model (or other permutation-dependent causal models), in comparison to the Gumbel model?**
>
>
> I will revisit my decision during the reviewer discussion period, but at the moment, I still feel that this paper is not ready for publication. It has promise, and the authors clearly have a compelling vision. So it would be a shame to publish this paper prematurely, i.e., before the vision is soundly supported by the science.
>
>
> FURTHER COMMENTS
>
>  - Definition C.1 (specifically, equation 9) does not typecheck; the right hand side is a random variable, not a probability distribution.
>
>  - change spacing or add parens in eqn (3)

---

> ### Comment · Reviewer_F4Ae · 2024-11-26
> **concrete possibilities for causal formalism**
>
> Here are some options, ordered as I would prefer them:
>
> 1. Consider an infinite family $\mathcal M_1, \mathcal M_2, \ldots$ of truncated SEMs. If the generated string has $n$ tokens, then the SEMs  {$\{ \mathcal M_i : i < n\}$} in this family that are truncated at length $i < n$ will not generate an EOS token, and all SEMs truncated at $i > n$ will agree on the generated string including the EOS token.
>
> 2. Fix a maximum length $M$ of sentence.  This is actually quite a reasonable assumption; in practice, we are not interested in sentences that have more than $10^{30}$ words, for instance. I personally do not find think the theoretical contribution of this work is substantially richer for considering sentences of unbounded length.  You can always mention that the restriction is artificial, and interventions, etc. can be extended to the infinite case as well.
>
> 3. Stick with GSEMs. Put the details (including the definition itself) in the appendix; what's important is just that it allows for infinite variables. State that you're defining interventions in the normal way, and hence not making use of most of the "generalized" part of GSEMs.

---

> > ### Author Response · Authors · 2024-11-26
> > **thanks!**
> >
> > We considered that approach. What we are opting for now is this. We will define something that is much less expressive than a GSEM -- basically, an infinite acyclic causal DAG where we have a well-founded partial order. We can inductively show (trivially) that such a structure entails a unique distribution. Importantly, we can also define interventions as one normally does with SEMs. This avoids the influence of truncation but also avoids the sledge hammer of the GSEM definition.
> >
> > Thanks again for engaging with this paper. It really made our work better!

---

> > > ### Comment · Reviewer_F4Ae · 2024-11-26
> > >
> > > Defining your own generalization of an ayclic causal model is also a reasonable approach, although I worry that doing it precisely will require more than the alternatives.  (I don't see what you mean by "influence" of truncation; I believe what I sketched above is a simple way of exactly capturing what you are interested in. But you are of course welcome to do something different.)
> > >
> > > I look forward to the final draft.

---

> ### Author Response · Authors · 2024-11-28
>
> Dear Reviewer F4Ae,
>
> We have updated the submission that addresses the discussed concerns (the important changes are now marked in blue).
> In particular, we highlight the following modifications:
> 1. We thought deeply about your suggestion about truncation. And, we want to emphasize again how much we appreciate the time you have put into making a detailed technical suggestion. However, we opted to go another way (as mentioned in our past comment) in removing the GSEM definition, as suggested by you in a previous comment. Our revised manuscript offers a definition of a well-founded SEM (generalizing an acyclic SEM) that naturally serves our purposes. We can think abut this definition as just the relevant parts of the GSEM definition without the added overhead. It simplifies our exposition of interventions, but also is sufficient to guarantee a unique solution, for which we cite Halpern and Peters (2021), rather than quoting their definition. We hope this alleviates the concerns.
> 2. The manuscript now contains significant hedging in the introduction and an improved title. Additionally, we have included a lively discussion about identifiability. While we have sharpened our arguments for why a Gumbel distribution is a useful choice (and a natural first step), we also discuss other options. Indeed, we believe the simplicity of the Gumbel-max parameterization of the softmax paves the road for future inquiry into the right (or at least a good) causal structure behind a language model—we believe this could be a valuable contribution of the added content. We hope this clarifies our objective and makes the contributions more in line with the motivation.
> 3. We have also rewritten the sampling algorithm in more abstract pseudocode (Alg. 1) and provided a more precise proposition (Prop. 3.1) that supports it. We have also included a detailed proof in the appendix that references the work cited in the previous versions.
> 4. As suggested, we have included a paragraph explaining the experimental setup in more detail.
> 5. The final version also addresses some minor concerns, such as defining a function that directly computes the logits defined by a language encoder, $\mathbf{E}$, and $\mathbf{b}$, as well as more careful treatment of random variables and their outcomes.
>
> We hope the modifications address your concerns. We would also like to sincerely thank you for the time and effort invested in your engagement with the work; we believe your feedback resulted in a much better manuscript! We look forward to hearing your final thoughts on the draft.

---

> ### Author Response · Authors · 2024-12-02
> **follow-up**
>
> Dear Reviewer F4Ae,
>
> We hesitate to write again given how much time you have invested in our paper. However, we wanted to ever so gently nudge you to see if our final draft manages to assuage your concerns. Thanks again for making this a very helpful review process for us.

---

> > ### Comment · Reviewer_F4Ae · 2024-12-02
> >
> > I have flipped through your newest version, and find that it is indeed significantly better than the previous ones.
> > I think I am very likely to increase my score when I get a chance to review it again carefully.
> > I do not see any more major issues at the moment.

---

### Official Review · Reviewer_tmKE · 2024-11-06

**Soundness:** 3
**Presentation:** 2
**Contribution:** 3
**Rating:** 6
**Confidence:** 4

**Summary:**

The paper first reframes the decoder-only language models as Generalized Structural Equation Models (GSEMs) with the Gumbel-max trick. Building on this framework, the authors propose an algorithm leveraging hindsight Gumbel sampling to infer and fix latent exogenous variables, enabling the generation of counterfactual versions of observed strings. Experimental results demonstrate that this method produces meaningful counterfactual pairs and further uncovers significant limitations of existing intervention techniques.

**Strengths:**

The paper introduces a novel framework that reframes language models through the lens of causal inference, which offers a new perspective for analyzing the causal mechanisms within language models. Building on this new framework, the paper also proposes a novel approach to generate counterfactual strings for observed strings. Interestingly, the proposed method provides a systematic means to evaluate existing intervention techniques, revealing previously unrecognized side effects.

**Weaknesses:**

1. The presentation of the paper could benefit from clearer emphasis on its primary contribution: generating counterfactual pairs. It would be helpful if the authors clarified early on that the intervention models employed are drawn from existing methods. Before reaching Section 4, there may be some ambiguity regarding how to obtain the counterfactual encoder $\tilde{h}$, which seems the hardest part for counterfactual generations.
2. The way to find $U_t$ should be refined:
     - Given the counterfactual string sampling method in Corollary 3.1, infinitely many vectors $U_t$ could theoretically be sampled, potentially resulting in different counterfactual generations. The paper currently presents only one counterfactual string per original string, and it would be beneficial to provide additional evidence demonstrating consistency across different samples of $U_t$.
     - While the Hindsight Gumbel Sampling can generate $U_t$ to ensure the next token is $\bar{w_t}$, there is limited theoretical or empirical discussion supporting this choice. Specifically, since $U_t(\bar{w_t})$ satisfies $\pi(\bar{w_t})+U_t(\bar{w_t})=\max_{\bar{w}\in\bar{\Sigma}}\pi(\bar{w})+U_t(\bar{w})$, the marginal distribution of $U_t(\bar{w_t})$ shouldn't be Gumbel(0,1) as mentioned in Proposition 3.1.
3. Additional experiments would enhance the persuasiveness of the results. For instance, demonstrating counterfactual generations across multiple Hindsight Gumbel Sampling trials would reinforce the approach’s robustness. Further, examining counterfactuals of counterfactual strings and comparing them to the original strings could offer valuable insights into the reliability and utility of the proposed method.

**Questions:**

See weakness.

---

> ### Author Response · Authors · 2024-11-14
> **Response**
>
> We thank the reviewer for their thoughtful and detailed feedback. We appreciate the recognition of our novel framework and the systematic evaluation of intervention techniques. We also value the suggestions provided to improve the clarity and empirical validation of our method. We address the concerns and questions raised as follows:
>
> 1. **Justification of Proposition 3.1 and Hindsight Gumbel Sampling**
>
> The reviewer questions the theoretical foundation of our Hindsight Gumbel Sampling algorithm, specifically challenging the independence of the sampled Gumbel noise and suggesting that the marginal distribution of  $U^t$ given an observed sentence might not remain \(\text{Gumbel}(0, 1)\). We want to explain the key justification provided in Proposition 3.1.
>
> The central insight of Proposition 3.1 is that the Gumbel-max trick allows us to *exactly* recover the posterior distribution \$P(U \mid \text{observed sentence})$ by leveraging the independence property of the Gumbel noise. The Gumbel-max trick ensures that the noise $U^t(w)$ for the sampled token $w$ (i.e., the observed token) can in fact be modeled as $\text{Gumbel}(0, 1)$. This follows from the key property of the Gumbel distribution, the independence between the max value and the rest, which allows us to sample from the posterior by first sampling a *standard* gumbel and then sampling the rest of the noise from the truncated distribution.
>
> By conditioning on the observed outcome (i.e., the chosen token), we ensure that $U^t(w_t) + \pi(w_t) \geq U^t(w) + \pi(w)$ for all  $w \neq w_t $. This condition precisely defines a truncated Gumbel distribution for the remaining noise variables. Thus, our method correctly samples from the posterior distribution of $U$. We emphasize that this approach is grounded in well-established properties of the Gumbel-max trick, and the independence of the Gumbel noise is key to its validity. We refer to the proof for this property in the appendix.
>
> 2.  ***Clarifying the primary contribution and the counterfactual encoder***
>
> We appreciate the feedback regarding the presentation of our primary contribution. While the introduction outlines our reframing of language models through causal inference and the novel counterfactual generation method, we will revise the early sections to provide a clearer and more explicit emphasis on the counterfactual generation as a core contribution.
> We agree that the description of obtaining the counterfactual encoder could be more explicit. In our experiments, we use existing intervention methods to define the counterfactual encoder (e.g., MEMIT, linear steering). We will make this clearer in the revised draft and emphasize that our contribution lies in the generation of true counterfactual strings and the analysis framework, not in proposing new intervention methods.
>
> 3. ***Addressing concerns on sampling variability***
>
> The reviewer pointed out that in our evaluation, we sample a single counterfactual, whereas we actually have a distribution over counterfactuals. This observation is valid. However, since our results are negative (i.e., we demonstrate that the counterfactuals tend to deviate significantly from the originals), using a single Monte Carlo sample, averaged over hundreds of different texts, is sufficient to reveal the differences between the originals and the counterfactuals. It is unlikely that we would consistently observe such a difference with just one sample but not with a larger number of samples (N >> 1). Nonetheless, to address this concern empirically, we conducted an additional experiment where we generated 10 counterfactuals per original sentence for the mimic-gender interventions. The results show a high mean cosine similarity of 0.88 between each group of 10 counterfactuals under the E5 model, indicating a strong degree of semantic similarity. Furthermore, we repeat the analysis of the normalized length of the longest prefix between the original and the counterfactuals. The median length changes from 0.281 (for N=1) to 0.265 (for N=10). We will repeat all experiments with N=10 for the final version of this work.
>
> Finally, we appreciate the suggestion to explore counterfactuals of counterfactuals as an additional robustness check. We agree that this could provide deeper insights into the reliability and utility of our method. We will include such experiments in the final version of this work.

---

> > ### Comment · Reviewer_tmKE · 2024-11-22
> >
> > Thanks for your detailed explanation. I raised my score.

---

> > > ### Author Response · Authors · 2024-11-22
> > > **Response**
> > >
> > > Thank you!

---

### Author Response · Authors · 2024-11-22
**General response**

We sincerely thank all the reviewers for their thoughtful feedback. Attached is a revised PDF that incorporates some of our updates based on the feedback provided. Notably, we have added a rigorous discussion on the identifiability of the GSEM construction (in a new section at the appendix), refined the presentation of the construction itself, and improved the notation in several places based on your suggestions. For ease of reference, the main changes are highlighted in red (both in the main text, and in the appendix). Additionally, we have provided individual responses to each reviewer.  Several reviewers (JN9h,tmKE) asked about the sampling mechanism, which seemed unjustified. In the individual comments, we explain that it is based on the independence properties of the Gumebl distribution, and refer to the proof.

We would greatly appreciate your feedback on whether our responses adequately address all concerns.

---

### Author Response · Authors · 2024-11-28
**Final Draft Uploaded**

Dear Reviewers,

We sincerely appreciate the extensive feedback and constructive suggestions provided during the review process. We believe the comments have improved us in refining the manuscript. We have uploaded its final version. In it, we have focused on clarifying the experimental setup and providing a more robust justification for the hindsight sampling methodology, as well as a more thorough discussion of the exact sampling scheme that we apply. Additionally, we have adopted a new simpler causal model, which we believe enhances the clarity and accessibility of our framework.

Thank you again for your valuable input. We hope that the updated manuscript aligns more closely with your expectations and addresses the points raised during the review process.

---

### Meta-Review · Area_Chair_vgCL · 2024-12-21

**Metareview:**

The authors propose modeling a language model output as an SCM with Gumbel exogenous noise. They use this model to perform counterfactual sampling. The proposed method does not give the ground truth counterfactuals as the paper was originally named, which was misleading. The authors changed the title to more suitable Gumber Counterfactuals based on reviewer input. There is also some difference in posterior sampling of the noise terms compared to standard abduction step where authors get one specific sample, per another reviewer's feedback. The side-by-side comparison between ground truth counterfactuals, which would require knowing the model would be useful. Indeed the ground truth model in a language modeling context is not even clear. I believe the proposed method should be more closely positioned as "how does the pretrained LMs react to Paerl's 3-steps of counterfactual sampling, if we adopted a Gumbel noise model".

Additional comments based on my reading:

Not sure if Pearl's 88 book is the right reference for causal calculus. I recommend authors check Pearl 1995 instead.

"Structural equation modeling" seems to be used as a proxy for structural causal models. Please use the right notion. The former typically refers to the linear setting. So the authors should use SCM not SEM to be precise.

"The exact mechanism is not of importance when one is concerned with correlational or interventional questions" This claim, without context is misleading. Please provide the necessary context. I believe the authors mean in the graph given above these are straightforward ID from OBS. Not true in general.

In experiments, "ground truth counterfactual" is assumed to come from some previous techniques that "intervene" on the model. Despite authors making the clear distinction between interventions (layer 2) and counterfactual (layer 3) in the earlier parts of the paper, they avoid this in experiments. Please explain why you think these "interventional" samples can be used as valid counterfactual ground truth samples.

**Additional Comments On Reviewer Discussion:**

Many reviewers engaged with the authors and changed their scores to accept after clarifications and edits. There was not much discussion afterwards. Despite the positive score, I believe this paper remains to be borderline leaning towards accept based on my own reading and the contents of the reviews instead of the assigned points.

---

### Decision · Program_Chairs · 2025-01-22

Accept (Poster)